# Bayesian modeling reveals metabolite-dependent ultrasensitivity in the cyanobacterial circadian clock

Lu Hong[1] (iD), Danylo O Lavrentovich[2,†] (iD), Archana Chavan[3], Eugene Leypunskiy[1] (iD), Eileen Li[4], Charles Matthews[4,‡], Andy LiWang[3,5,6,7,8,9] (iD), Michael J Rust[10,11,12,*] & Aaron R Dinner[2,11,13,**] (iD)

## Abstract

Mathematical models can enable a predictive understanding of mechanism in cell biology by quantitatively describing complex networks of interactions, but such models are often poorly constrained by available data. Owing to its relative biochemical simplicity, the core circadian oscillator in *Synechococcus elongatus* has become a prototypical system for studying how collective dynamics emerge from molecular interactions. The oscillator consists of only three proteins, KaiA, KaiB, and KaiC, and near-24-h cycles of KaiC phosphorylation can be reconstituted *in vitro*. Here, we formulate a molecularly detailed but mechanistically naive model of the KaiA—KaiC subsystem and fit it directly to experimental data within a Bayesian parameter estimation framework. Analysis of the fits consistently reveals an ultrasensitive response for KaiC phosphorylation as a function of KaiA concentration, which we confirm experimentally. This ultrasensitivity primarily results from the differential affinity of KaiA for competing nucleotide-bound states of KaiC. We argue that the ultrasensitive stimulus–response relation likely plays an important role in metabolic compensation by suppressing premature phosphorylation at nighttime.

**Keywords** Bayes factor; kinetic modeling; Markov chain Monte Carlo; robustness; substrate competition

**Subject Categories** Computational Biology; Metabolism; Signal Transduction

**Mol Syst Biol. (2020) 16: e9355**

## Introduction

Achieving a predictive understanding of biological systems and chemical reaction networks is challenging because complex behavior can emerge from even a small number of interacting components. Classic examples include the propagation of action potentials in neurobiology and chemical oscillators such as the Belousov–Zhabotinsky reaction. The collective dynamics in such systems cannot be easily intuited through qualitative reasoning alone, and thus, mathematical modeling plays an important role in summarizing and interpreting existing observations and formulating testable, quantitative hypotheses. However, it can be difficult to rationalize how collective dynamics emerge from specific molecular features. One approach to addressing this issue is to compare mathematical representations of competing molecular mechanisms based on their abilities to fit experimental data.

The circadian clock from the cyanobacterium *Synechococcus elongatus* (Johnson *et al*, 2011) represents a unique opportunity to use model fitting to learn biochemical mechanisms. This is because the core oscillator can be reconstituted in a test tube from a small number of components. The simplicity of the system makes it possible to both cleanly model the basic biochemical

1 Graduate Program in Biophysical Sciences, University of Chicago, Chicago, IL, USA
2 Department of Chemistry, University of Chicago, Chicago, IL, USA
3 School of Natural Sciences, University of California, Merced, CA, USA
4 Department of Statistics, University of Chicago, Chicago, IL, USA
5 Quantitative and Systems Biology, University of California, Merced, CA, USA
6 Center for Circadian Biology, University of California, San Diego, La Jolla, CA, USA
7 Chemistry and Chemical Biology, University of California, Merced, CA, USA
8 Health Sciences Research Institute, University of California, Merced, CA, USA
9 Center for Cellular and Biomolecular Machines, University of California, Merced, CA, USA
10 Department of Molecular Genetics and Cell Biology, University of Chicago, Chicago, IL, USA
11 Institute for Biophysical Dynamics, University of Chicago, Chicago, IL, USA
12 Institute for Genomics and Systems Biology, University of Chicago, Chicago, IL, USA
13 James Franck Institute, University of Chicago, Chicago, IL, USA
  *Corresponding author. Tel: +1 773 834 1463; E-mail: mrust@uchicago.edu
  **Corresponding author. Tel: +1 773 702 2330; E-mail: dinner@uchicago.edu
  †Present address: Department of Organismic and Evolutionary Biology, Harvard University, Cambridge, MA, USA
  ‡Present address: School of Mathematics, University of Edinburgh, Edinburgh, UK

events in the circadian cycle and to collect quantitative data under well-controlled conditions. The core oscillator of *S. elongatus* consists of three proteins, KaiA, KaiB, and KaiC, which self-organize to generate a near-24-h rhythm in KaiC phosphorylation. The basic biochemical events are well-established (Swan *et al*, 2018). KaiC is an ATPase (Terauchi *et al*, 2007) that phosphorylates and dephosphorylates itself by transfer of phosphoryl groups from and to bound nucleotides (Egli *et al*, 2012; Nishiwaki & Kondo, 2012); KaiA-dependent nucleotide exchange reactions drive the phosphorylation phase of the cycle (Nishiwaki-Ohkawa *et al*, 2014), and KaiB-mediated sequestration of KaiA leads to dephosphorylation. The reconstituted oscillator retains many of the hallmarks of circadian rhythms in living organisms (Nakajima *et al*, 2005; Yoshida *et al*, 2009; Rust *et al*, 2011; Leypunskiy *et al*, 2017).

Yet, questions remain about how the clock couples to environmental conditions while maintaining a robust ~ 24-h rhythm (Yoshida *et al*, 2009; Phong *et al*, 2013; Leypunskiy *et al*, 2017; Murayama *et al*, 2017). The Kai oscillator senses changes in the relative concentrations of ATP to ADP in solution, which allows entrainment to metabolic rhythms (Rust *et al*, 2011; Phong *et al*, 2013; Leypunskiy *et al*, 2017). KaiA modulates these dynamics via its function as a nucleotide exchange factor, but how the system adapts and responds to changes in metabolic conditions is not clear. Models that account for the possible protein complexes, including the interplay of nucleotide-bound and phosphorylation states, can be dauntingly complex (e.g., Lin *et al*, 2014; Paijmans *et al*, 2017b). How to fit them to data and interpret the results is an area of active research.

Here, we use a data-driven Bayesian approach to estimate the parameters of a molecularly detailed kinetic model of the KaiA–KaiC subsystem, with the goal of learning the features required to capture the behavior of the system during the phosphorylation phase of the clock cycle quantitatively (Fig 1A). The model describes the coupling between KaiA, nucleotides (ATP and ADP) in solution, KaiC phosphorylation, and KaiC nucleotide-bound states. To provide training data for the model, we collected kinetic time series measuring KaiC phosphorylation kinetics over a wide range of KaiA concentrations ([KaiA]) and %ATP (defined as 100%[ATP]/([ATP] + [ADP])). Although such data do not give us direct access to all relevant states of the KaiA–KaiC subsystem, they place constraints on the underlying molecular interactions. Bayesian statistics (Wasserman, 2000; MacKay & Kay, 2003) have found diverse applications in systems biology (Flaherty *et al*, 2008; Klinke,

2009; Toni *et al*, 2009; Xu *et al*, 2010; Schmidl *et al*, 2012; Eydgahi *et al*, 2013; Pullen & Morris, 2014; Mello *et al*, 2018), including circadian biology (Higham & Husmeier, 2013; Trejo Banos *et al*, 2015; Martins *et al*, 2018). Here, they provide a unified framework for estimating parameter values, quantifying the importance of specific model elements, and making mechanistic predictions from the model.

By Markov chain Monte Carlo (MCMC) sampling, we obtain an ensemble of parameter sets that fit the data. Even with extensive training data, many microscopic parameters in the model are not tightly constrained. Despite this, we show that this ensemble of fits robustly makes predictions that are borne out in experimental tests (Brown & Sethna, 2003; Gutenkunst *et al*, 2007). In particular, the model reveals a previously unappreciated ultrasensitive dependence of phosphorylation on the concentration of KaiA, with strong nonlinearity at low [KaiA], conditions that likely apply near the nighttime to daytime transition point, when a large fraction of KaiA molecules are inhibited. Importantly, we find that the threshold KaiA concentration varies with the %ATP in solution. This ultrasensitive response primarily arises from a differential affinity of KaiA for different nucleotide-bound states of KaiC. This mechanism is analogous to substrate competition (Ferrell & Ha, 2014b), where kinetic competition of multiple enzyme substrates leads to ultrasensitivity.

Lastly, we consider the implications of these results for the full oscillator, in which KaiC rhythmically switches between phosphorylation and dephosphorylation. A well-known mechanism for the inhibition of KaiA is its sequestration into KaiBC complexes (Chang *et al*, 2012), which form when KaiC is sufficiently phosphorylated (Rust *et al*, 2007); this mechanism serves as a delayed negative feedback loop in the oscillator. The KaiB-independent thresholding phenomenon we describe here is strongest when KaiC phosphorylation is low and when KaiC is ADP-bound. This suggests there are at least two mechanisms that work together during the cycle to prevent KaiA from acting at inappropriate times, and that the relative strength of the two mechanisms varies with the nucleotide pool.

Incorporation of the ultrasensitive response to KaiA into a mathematical model of the full oscillator suggests that this effect both stabilizes the period against changes in the nucleotide pool and allows oscillations to persist even when KaiB binds KaiA relatively weakly. Consistent with this prediction, we find that a substantial amount of KaiA is not bound by KaiB even when the clock is

---

**Figure 1. Phosphorylation data are fit by a mechanistically naive kinetic model.**

A   An outline of the data-driven Bayesian model fitting approach employed in this work.

B   To constrain the model, measurements of KaiC phosphorylation kinetics were collected at six [KaiA] and three %ATP conditions. The curves represent the best fit model prediction.

C   A schematic of the mass-action kinetics model. The model elaborates on the autophosphorylation reactions of KaiC by explicitly keeping track of the time evolution of the KaiC phosphoforms and nucleotide-bound states; conversions among these states are mediated by phosphotransfer, nucleotide exchange, ATP hydrolysis, and KaiA (un)binding. Note that the KaiA binding reactions are second-order, but KaiA concentration ([A]) is written as part of the effective first-order rate constant. See the main text for a discussion of the state and rate constant nomenclature and Fig EV1A for a schematic of the full model.

D   The posterior distributions for log KaiA dissociation constants (base 10). The horizontal axis represents the affinity for ADP-bound KaiC, and the vertical axis represents the affinity for ATP-bound KaiC; $X \in \{U, T, S, D\}$ and corresponds to the four colors of the KaiC phosphoforms, as in panel C. The asterisks represent the best fit, and the contour lines represent the 95% and 68% highest posterior density regions (HDR). The dashed line represents the $K_d^{X,TP} = K_d^{X,DP}$ line, so that densities above the line indicate higher affinity for the ADP-bound states and densities below the line indicate higher affinity for the ATP-bound states.

Source data are available online for this figure.

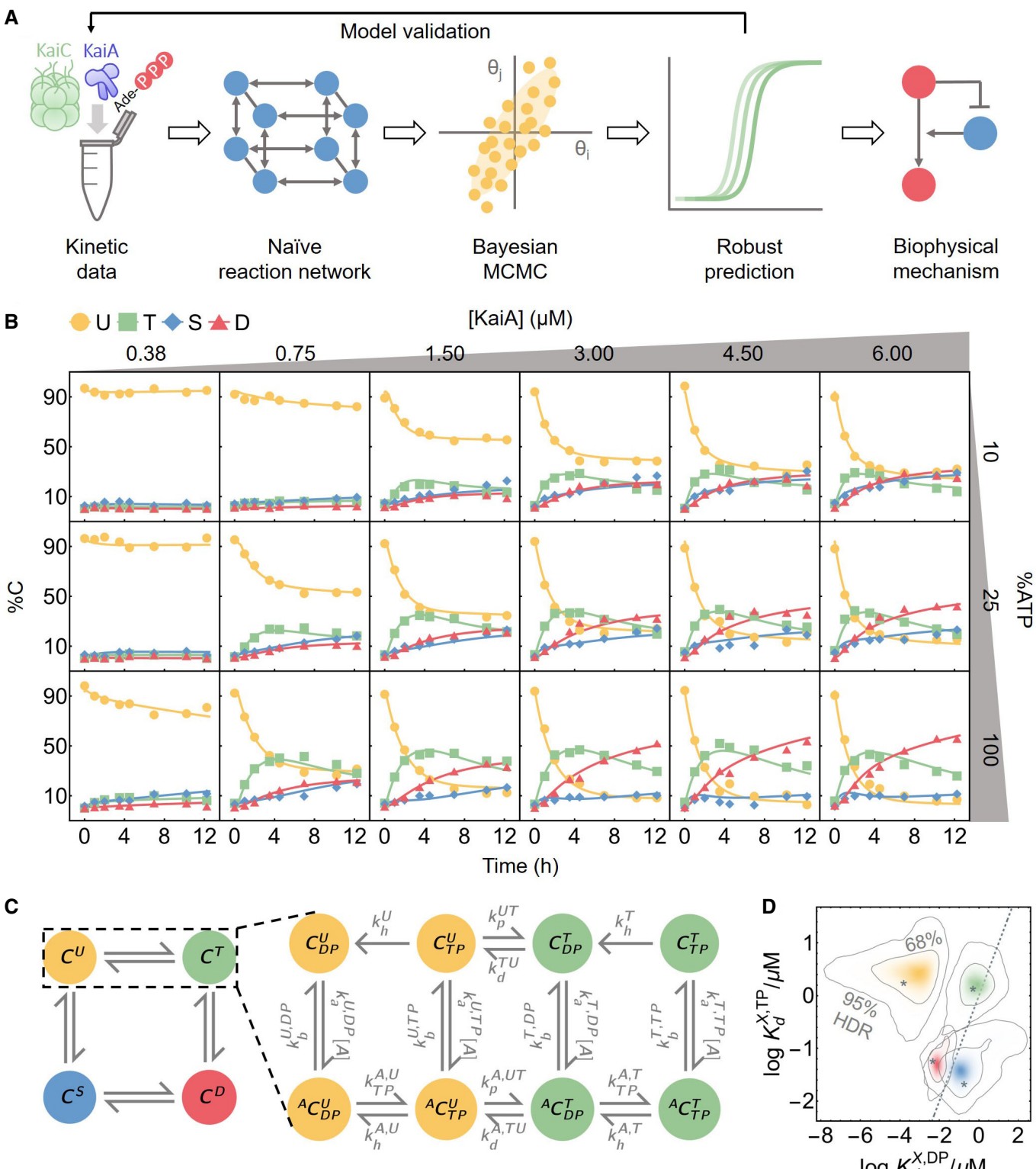

**Figure 1.**

dephosphorylating. These results shed new light on metabolic compensation, a property that allows robust 24-h oscillation in spite of changes in %ATP conditions (Johnson & Egli, 2014). Taken together, our results show how the Bayesian framework combined with extensive training data can be used to discover unanticipated mechanisms and direct experimental investigations.

# Results

### A molecular model of KaiA—KaiC dynamics

To probe the response of KaiC phosphorylation to a wide range of metabolic conditions, we made kinetic measurements of KaiC phosphorylation at three %ATP conditions and six [KaiA] conditions while holding the KaiC concentration constant (Fig 1B). KaiC is a homohexamer and each subunit has two duplicated ATPase domains, termed CI and CII (Hayashi *et al*, 2003; Pattanayek *et al*, 2004; Terauchi *et al*, 2007). The CII domain can autophosphorylate via a bidirectional phosphotransferase mechanism (Egli *et al*, 2012; Nishiwaki & Kondo, 2012) with two phosphorylation sites. Each KaiC subunit thus has four phosphoforms: the unphosphorylated (U), phosphoserine-431 (S), phosphothreonine-432 (T), and doubly phosphorylated (D) states, each of which peaks at a distinct time in the oscillation (Xu *et al*, 2004; Nishiwaki *et al*, 2007; Rust *et al*, 2007).

Our strategy is to fit these data with a model of the KaiC catalytic cycle in CII with a minimum of simplifying assumptions. To this end, we formulate a model based on mass-action kinetics. We explicitly keep track of three properties of the CII domain of each KaiC subunit: its phosphorylation status (right superscripts in Fig 1C), nucleotide-bound state (right subscript), and whether or not KaiA is bound (left superscript). We do not consider CI or the hexameric nature of KaiC explicitly (see Appendix for further discussion). There are thus 16 possible KaiC states, eight of which are shown in Fig 1C, along with the phosphotransfer, nucleotide exchange, KaiA (un)binding, and hydrolysis reactions that connect the states (see Fig EV1A for the full model structure). We also considered the possibility that nucleotides might interact directly with KaiA, which could allow KaiA's activity to directly depend on nucleotides in solution. However, we did not detect any direct interaction between KaiA and ATP or ADP using NMR spectroscopy (Appendix Fig S1), so we do not allow for this scenario in the model.

In the rest of this section, we further elaborate our model by stepping through the four classes of reactions that we include; additional details can be found in Materials and Methods. In the next section, we describe in qualitative terms the fitting procedure and analyze the extent to which the model was constrained by the training dataset. For readers who are primarily interested in the biochemical conclusions of the model, these results can be skipped without loss of continuity.

#### Phosphotransfer
The KaiC CII domain has reversible phosphotransferase activity (Egli *et al*, 2012; Nishiwaki & Kondo, 2012); it can transfer a γ-phosphate group from a bound ATP to a phosphorylation site, but unlike a typical phosphatase, it regenerates ATP from ADP during dephosphorylation, i.e.,

$$C_{TP}^{X} \underset{k_d}{\overset{k_p}{\rightleftharpoons}} C_{DP}^{Y} \tag{1}$$

where $(X,Y) \in \{(U,T),(U,S),(T,D),(S,D)\}$. This mechanism implies that the nucleotide-bound state of KaiC has a significant impact on the net direction of its phosphotransferase activity: An ATP-bound

KaiC presumably cannot dephosphorylate, and an ADP-bound KaiC cannot phosphorylate.

#### Nucleotide exchange
KaiA binding to the CII domain (Kim *et al*, 2008; Pattanayek & Egli, 2015) stimulates KaiC autophosphorylation (Iwasaki *et al*, 2002; Williams *et al*, 2002; Kageyama *et al*, 2006). Recent work has shown that KaiA can bind to KaiC and act as a nucleotide exchange factor (Nishiwaki-Ohkawa *et al*, 2014) by facilitating conformational changes at the subunit interface that promote solvent exposure of the nucleotide-binding pocket (Hong *et al*, 2018). It is currently unclear whether this nucleotide exchange activity is responsible for all of KaiA's effect on KaiC or whether it alters the KaiC catalytic cycle in other ways (see Appendix for further analysis of this issue). The reversible binding of KaiA

$$C \underset{k_b}{\overset{k_a[A]}{\rightleftharpoons}} {}^{A}C \tag{2}$$

contributes two classes of rate constants, $k_a$ and $k_b$.

Because the CII domain of KaiC releases its bound nucleotides very slowly in the absence of KaiA (Nishiwaki-Ohkawa *et al*, 2014), we ignore the possibility of KaiA-independent nucleotide exchange in the model. Under the assumptions that (i) the apo state is in a quasi-steady state, (ii) the ADP and ATP on rates are identical, and (iii) ATP release is slow, nucleotide exchange can be modeled as a one-step reaction:

$$ATP + {}^{A}C_{DP} \overset{k_{TP}^{A}}{\rightarrow} {}^{+A}C_{TP} + ADP \tag{3}$$

where

$$k_{TP}^{A} = k_r^{DP} \frac{[ATP]}{[ATP] + [ADP]} \tag{4}$$

and $k_r^{DP}$ is the ADP dissociation rate constant. Nucleotide exchange thus contributes one class of rate constant, $k_r^{DP}$. See Materials and Methods for the derivation of (4).

#### ATP hydrolysis
Finally, we allow for irreversible ATP hydrolysis in the CII domain

$$C_{TP} \overset{k_h}{\rightarrow} C_{DP} + P_i \tag{5}$$

which contributes one class of rate constants, $k_h$. This assumption is important because this pathway is required in the model for sustained dephosphorylation by allowing ADP-bound KaiC to accumulate in the absence of KaiA. Because each KaiC molecule consumes relatively little ATP on the timescale of the simulations (Terauchi *et al*, 2007), we assume the solution ATP and ADP concentrations are constant.

#### State-dependent rates
Given the six classes of rate constants, $k_p$, $k_d$, $k_a$, $k_b$, $k_r^{DP}$, and $k_h$, we make the model maximally general, or mechanistically naive, by allowing each rate constant to potentially depend on the specific molecular state involved in the reaction. For example, the KaiA dissociation rate constant is allowed to vary depending on the

nucleotide-bound state and phosphoform background of KaiC, and thus, the dissociation rate constants for the ADP-bound U ($k_b^{U,DP}$) and ATP-bound T phosphoforms ($k_b^{T,TP}$) are two independent model parameters. In this way, the parameter fitting and model comparison procedures automatically test specific biochemical hypotheses about the functions of KaiA and KaiC. For example, allowing the KaiA off rates to depend on the nucleotide-bound states is equivalent to the hypothesis that KaiA has different dwell times for ATP- vs. ADP-bound states of KaiC. In fact, because each reaction has an independent rate constant, except for thermodynamic constraints of detailed balance, the fitting procedure effectively allows for simultaneous testing of all possible two-way interactions of the three categories of KaiC properties, without *a priori* preference for any particular mechanism.

## The data constrain the parameters to widely varying degrees

We estimate the model parameters through a Bayesian framework. In this framework, we maximize the posterior probability, which is proportional to the product of the prior distribution and the likelihood function. Here, we interpret the prior as representing subjective beliefs on the model parameters before experimental inputs, while the likelihood function quantifies the goodness of fit. Bayesian parameter estimation reduces to least-squares fitting under the assumption of normally distributed residuals and uniform priors. In practice, we find that direct numerical optimization of the posterior usually results in fits that are trapped in low probability local maxima (Appendix Fig S2B). Thus, we instead draw parameters from the prior distribution and then use a heuristic combination of MCMC sampling and optimization (Powell's algorithm) to explore the parameter space. The MCMC method that we use (Goodman & Weare, 2010; Foreman-Mackey *et al*, 2013) efficiently searches the parameter space by simulating an ensemble of parameter sets in parallel; the spread of the ensemble reflects the geometry of the posterior distribution and is used to guide the directions of Monte Carlo moves. See Materials and Methods for a more mathematical treatment of the fitting procedure and comparison of different numerical optimization and sampling methods.

We use this approach to fit the phosphorylation data (Fig 1B) together with previously published data on dephosphorylation (Rust *et al*, 2011), ATP hydrolysis rate (Terauchi *et al*, 2007), and the KaiA dwell time for each KaiC phosphoform (Kageyama *et al*, 2006; Mori *et al*, 2018) (see Materials and Methods). Overall, the model achieves excellent agreement with the training data (Figs 1B and EV2A–C). In the following analyses, we refer to model predictions using the best fit parameter values and quantify the uncertainties using the posterior distribution (see Appendix for further discussion of the convergence of the simulations).

We find that certain parameters, such as the hydrolysis rates in the U and T phosphoforms and the KaiA off rates from the U phosphoform, are tightly constrained, while many others, mainly involving S and D phosphoforms, are less constrained, in the sense that their posterior distributions span multiple orders of magnitude, exhibit multimodality, or cannot be reproduced over multiple independent runs (Fig EV1B). Some parameters are highly correlated, and certain combinations of the parameters are much better constrained than the individual parameters. For example, the

posterior distributions for the KaiA binding affinities (Fig 1D) appear better constrained than the on/off rates (Appendix Fig S3B).

Taken together, these results are consistent with the notion that collective fits of multiparameter models are generally "sloppy", meaning that the sensitivities of different combinations of parameters can range over orders of magnitude with no obvious gaps in the spectrum (Brown & Sethna, 2003; Gutenkunst *et al*, 2007). As we will see, we can nonetheless make useful predictions using the ensemble of model parameters, because the model behavior is constrained along the stiffest directions of the posterior distribution. By contrast, direct parameter measurements need to be both complete and precise to achieve similar predictive validity (Gutenkunst *et al*, 2007). We further characterize the structure of the parameter space in Appendix and Appendix Fig S3.

## KaiC (de)phosphorylation goes through transient kinetic intermediates

In the model, we can break down the kinetics of KaiC phosphorylation reactions and interpret the underlying molecular events. Here, we consider the phosphorylation kinetics at a standard reaction condition (3.5 μM KaiC, 1.5 μM KaiA, 100% ATP; Fig 2A and B, solid curves); we examine the effect of varying [KaiA] and %ATP in the following sections.

At the beginning of the phosphorylation reaction, KaiC molecules are predominantly in the ADP-bound U state ($C_{DP}^U$), the end product of the dephosphorylation pathway in the absence of KaiA (Fig 2A). With the addition of abundant KaiA, the $C_{DP}^U$ state becomes rapidly depleted within the first 10 minutes of the reaction and enters the $C_{TP}^U$ state. Consistent with the kinetic ordering observed in the full oscillator, the $C_{TP}^U$ population is primarily converted into the T phosphoform over the S phosphoform. The mechanism underlying the preference for the T phosphoform is not well constrained by the data, but it appears to be the result of more than just a difference in the relative U → T and U → S phosphorylation rates; a sensitivity analysis shows that the ordering of phosphorylation is also dependent on KaiA (un)binding kinetics (see Appendix and Appendix Fig S4). The ADP- and KaiA-bound T phosphoform states are unstable kinetic intermediates, and the population accumulates at the $C_{TP}^T$ bottleneck for the first 4 h. As phosphorylation reaches completion, the T phosphoform is converted first into $^A C_{TP}^D$ through the unstable ADP-bound intermediates and then to the $C_{TP}^D$ state; the populations of the $^A C_{TP}^D$ and $C_{TP}^D$ states are comparable at steady state. We note here, however, that previous measurements indicate that ∼ 30% of CII nucleotide-binding pockets should be ADP-bound in the presence of KaiA at steady state (Nishiwaki-Ohkawa *et al*, 2014), which suggests that the stability of the ADP-bound form is systematically underestimated by the model fit.

During the phosphorylation reaction, the amount of free KaiA is initially transiently depleted due to association with the ADP-bound U phosphoform (Fig 2B). Afterward, KaiA primarily associates with the ATP-bound S and D phosphoforms as they appear, but does not bind to the T phosphoform strongly. Therefore, even in the absence of KaiB, not all KaiA is free during the phosphorylation phase, and the amount of free KaiA is predicted to depend on both the affinities of the nucleotide-bound states and the mixture of KaiC phosphorylation states (Fig 1D).

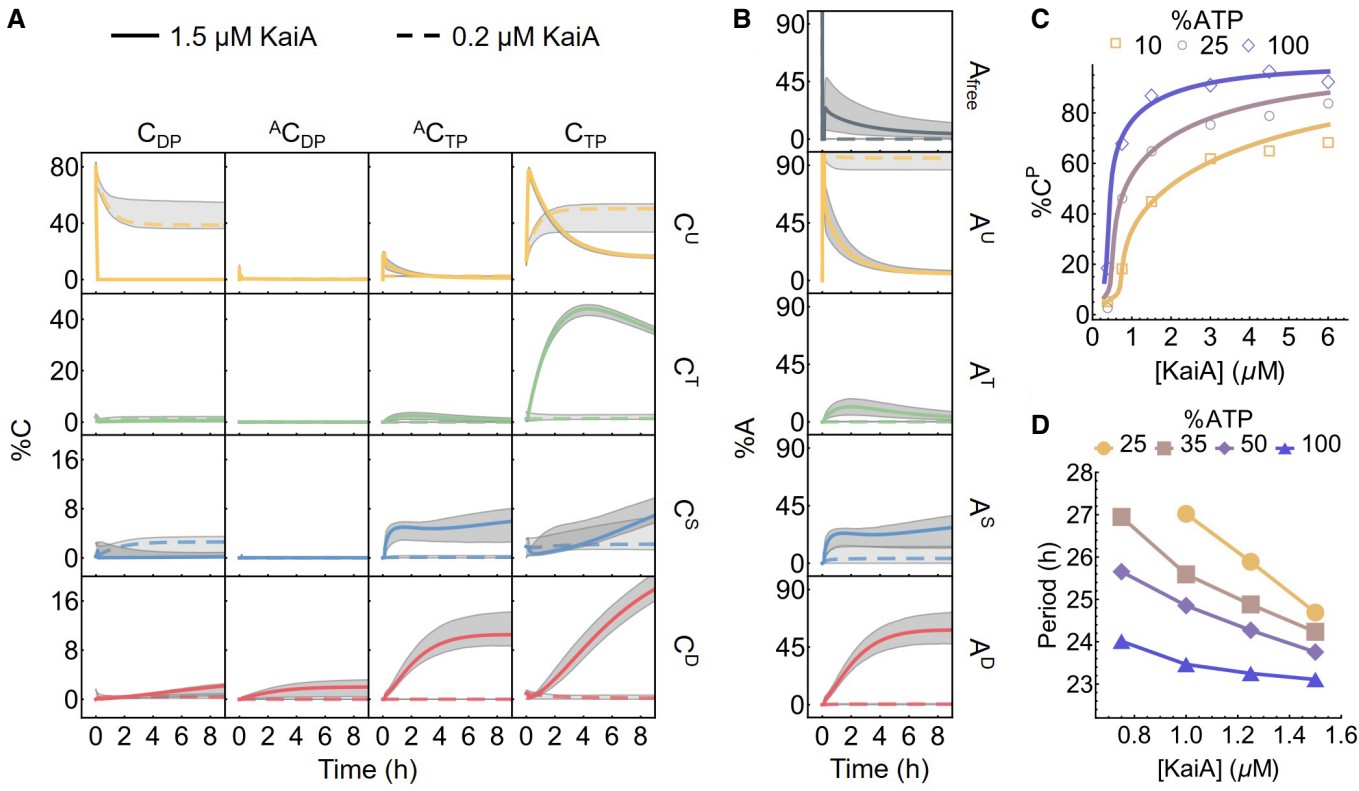

**Figure 2.  The model captures the kinetics of KaiC phosphorylation.**

A   The time evolution of all 16 KaiC states in a phosphorylation reaction with 100% ATP and either 1.5 μM (solid curves) or 0.2 μM (dashed curves) KaiA. The gray regions represent the 95% posterior interval. Refer to Fig EV1A for the KaiC state nomenclature.

B   The corresponding KaiA kinetics, broken down according to the phosphoform of the bound KaiC.

C   KaiA concentration can tune the sensitivity of the KaiC phosphorylation level to %ATP. The points represent the measured total percentage phosphorylation levels at t = 12.25 h (see Fig 1B for the full kinetics), and the curves represent the model prediction at the same time point.

D   KaiA concentration can tune the sensitivity of the clock period to %ATP. The period of the full KaiABC oscillator is calculated from fluorescence polarization measurements (see Fig EV3A and B for further analysis).

Source data are available online for this figure.

We were surprised to see that the model fit predicts that the KaiA binding affinity for the ATP-bound T phosphoform is lower than those for the S and D phosphoforms. This is apparently in contradiction with experimental results that show that S-phosphomimetic mutants reduce A-loop exposure and weaken KaiA binding, while T-phosphomimetic mutants have opposite effects (Chang *et al*, 2011; Tseng *et al*, 2014). Our model predicts that the nucleotide bound to CII has large impact on the affinity of KaiA for KaiC. One possibility is that phosphomimetic mutations may change the average nucleotide-bound state by altering the catalytic cycle, and that these nucleotide changes have a large effect on the experimental measurements. Future tests of this hypothesis through experiments that directly control for the bound nucleotide and models that represent both nucleotide- and phosphoform-dependent KaiA binding affinity will likely be able to separate the direct effect of phosphorylation from nucleotide effects.

The dephosphorylation pathway is simpler because KaiA is not involved. In the model, KaiC by itself has no nucleotide exchange activity in CII, and thus, phosphorylated KaiC molecules enter a cycle of dephosphorylation by the transfer of phosphoryl groups from the phosphorylation sites back to bound ADP molecules,

followed by ATP hydrolysis and removal of inorganic phosphate, until the protein reaches the $C_{DP}^{U}$ state (Fig EV2D). The ADP-bound forms of the T, S, and D phosphoforms are only transiently populated, suggesting that the dephosphorylation bottleneck is ATP hydrolysis, which makes bound ADP available as a cofactor for dephosphorylation, rather than the phosphotransfer itself. The kinetic preference for the D → S dephosphorylation pathway is the direct result of faster dephosphorylation via the D → S reaction compared to the D → T reaction (Fig EV1B; compare the posterior distribution of $k_{d}^{DS}$ with that of $k_{d}^{DT}$). During this process, KaiC can occasionally autophosphorylate, but it is driven irreversibly toward the dephosphorylated state by ATP hydrolysis. We note here that the independence of the dephosphorylation reaction from solution ADP (Rust *et al*, 2011) is a built-in feature of the model, since solution %ATP only affects the nucleotide exchange rate, which is assumed in the model to be zero in the absence of KaiA.

### KaiA concentration tunes clock sensitivity to %ATP

Our data and the model allow us to ask in detail how [KaiA] modulates the effect of %ATP on KaiC phosphorylation (Fig 2C).

Consistent with previous measurements (Rust *et al*, 2011; Phong *et al*, 2013), these results indicate that the near-steady-state ($t = 12.25$ h) total phosphorylation level of KaiC ($\%C^P = \%C^T + \%C^S + \%C^D$) is lower in the presence of ADP. Since we simultaneously vary %ATP and [KaiA], the data reveal that this inhibitory effect can be tuned by [KaiA]. In particular, the system is most insensitive to %ATP at either very low or very high [KaiA], while the %ATP sensitivity is the highest around [KaiA] = 0.75 μM.

Some %ATP sensitivity remains even at saturating KaiA concentrations (Fig 2C). This effect can be interpreted qualitatively in terms of the structure of the model. When there is more KaiA in solution, more KaiC goes through intermediate states that are in complex with KaiA, by Le Châtelier's principle. This shifts a larger fraction of the KaiC population to states that allow for exchange of bound ADP for ATP, which promotes phosphorylation. On the other hand, as the %ATP decreases, the ATP to ADP exchange rate decreases according to (4). When nucleotide exchange becomes less efficient, more KaiC stays in ADP-bound states, which are prone to dephosphorylation. In summary, [KaiA] and %ATP both act on the phosphorylation kinetics via the nucleotide exchange step. Solution %ATP directly regulates the exchange rate constant and sets its upper bound, while [KaiA] controls the population of exchange-competent KaiC and thus the effective exchange rate. Therefore, the effects of KaiA and increasing solution %ATP are not equivalent; because the effective exchange rate cannot exceed the limit set by %ATP, even a saturating amount of KaiA cannot fully compensate for low %ATP. In this sense, KaiA acts as an input regulator—high [KaiA] blunts the intrinsic sensitivity of KaiC to ADP in solution.

Given that the metabolic sensitivity of the KaiA—KaiC subsystem can be tuned by KaiA concentration, we asked whether metabolic sensitivity of the full oscillator period may also be tuned by KaiA. Such a scenario might allow the [KaiA]/[KaiC] ratio to be an important parameter *in vivo* for adjusting the sensitivity of the clock to the daily metabolic rhythm of the cell. To address this question, we characterized the dependence of the period of the *in vitro* KaiABC oscillator on [KaiA] and %ATP using a fluorescence polarization assay (Leypunskiy *et al*, 2017; Heisler *et al*, 2019) (Figs 2D and EV3A and B). Consistent with the hypothesis, we found that low KaiA concentration enhances the period sensitivity to %ATP compared to the standard condition (1.5 μM KaiA). These results suggest that the KaiA activity, and how it is controlled, plays a critical role in determining how responsive the oscillator is to metabolic changes.

## KaiC phosphorylation exhibits ultrasensitive dependence on KaiA levels

In addition to inferring kinetics of states not easily accessible to experiments, the model allows us to interpolate between the training data points and study the relation between KaiC phosphorylation, [KaiA], and %ATP at a much finer resolution. This analysis shows an ultrasensitive dependence of the steady-state $\%C^P$ on KaiA concentration (Fig 3A and D left). Specifically, we see a threshold–hyperbolic stimulus–response relation (Gomez-Uribe *et al*, 2007; Ferrell & Ha, 2014a), where KaiC phosphorylation is highly suppressed near the sub-micromolar [KaiA] regime, but then follows a right-shifted hyperbolic stimulus–response function once [KaiA] exceeds a threshold. Importantly, the threshold

depends on %ATP. The model makes similar predictions for the steady-state T, S, and D phosphoforms as well (Fig EV4A). However, because of the T → D and S → D phosphotransfer reactions, the stimulus–response relations of T and S are not monotonic functions of [KaiA] because high [KaiA] and high %ATP conditions stabilize the D phosphoform at the expense of the T and S phosphoforms.

Previous studies of KaiA—KaiC interactions examined the response of KaiC at relatively high KaiA concentrations (≥ 1.2 μM), comparable to the total amount of KaiA used in an oscillating reaction. Ma and Ranganathan (2012) investigated the steady-state stimulus–response relation, but did not consider the effect of %ATP or fully characterize the low [KaiA] regime. Previous reports of initial phosphorylation rates suggest that they exhibit a hyperbolic dependence on [KaiA] (Rust *et al*, 2007; Lin *et al*, 2014), similar to simple Michaelis–Menten enzyme–substrate systems. However, this does not imply that the steady-state stimulus–response relation is hyperbolic as well.

To assess the robustness of the model prediction of ultrasensitivity across the ensemble, we use two metrics proposed by Gunawardena (2005) to quantify the shape of the predicted stimulus–response curves for $\%C^P$ at any fixed %ATP: We use EC10 to measure the extent to which the curve acts as a threshold and EC90–EC10 to measure the extent to which the curve acts as a switch. Here, EC*x* is the KaiA concentration required to reach *x*% of the steady-state phosphorylation level at saturation. Figure 3E shows the distribution of these quantities in the ensemble at 25% ATP. Overall, these statistics are tightly constrained by the training data set and are clearly distinct from those from hyperbolic stimulus–response relations (Fig 3E, dashed gray line).

We then sought to experimentally determine the shape of the stimulus–response function. We measured KaiC phosphorylation at $t = 24$ h at various concentrations of [KaiA] at three %ATP conditions (Figs 3B and EV4B). Consistent with the model predictions, the experimentally derived stimulus–response relations are ultrasensitive with a %ATP-dependent phosphorylation threshold, and the stimulus–response relation of the S phosphoform at 100% ATP is non-monotonic. We then quantified the shape of the stimulus–response curve for $\%C^P$ at 25% ATP using the same two metrics defined above (Fig 3E, yellow star). At 25% ATP, the shape of the experimentally derived stimulus–response curve is close to that of the model prediction, but the model fit is systematically less threshold-like (i.e., smaller EC10) and less switch-like (i.e., larger EC90–EC10). This inconsistency is likely due to a combination of training data under-determining the shape of the curve at the sub-micromolar range (compare Figs 2C with 3B) and the fitting method under-estimating uncertainties (see Appendix).

Lastly, the saturating phosphorylation levels in the steady-state measurements appear systematically lower than those implied by the training data set (compare Fig 3B with D left). This may be a result of batch-to-batch variations in protein and nucleotide quantification. This difference can be corrected by refitting the model to the steady-state measurement (Figs 3B and EV4C). The refit results suggest that errors in protein and nucleotide concentrations primarily affect the kinetic properties of the S phosphoform in the model (Fig EV4D), but the refitting does not change the qualitative conclusions.

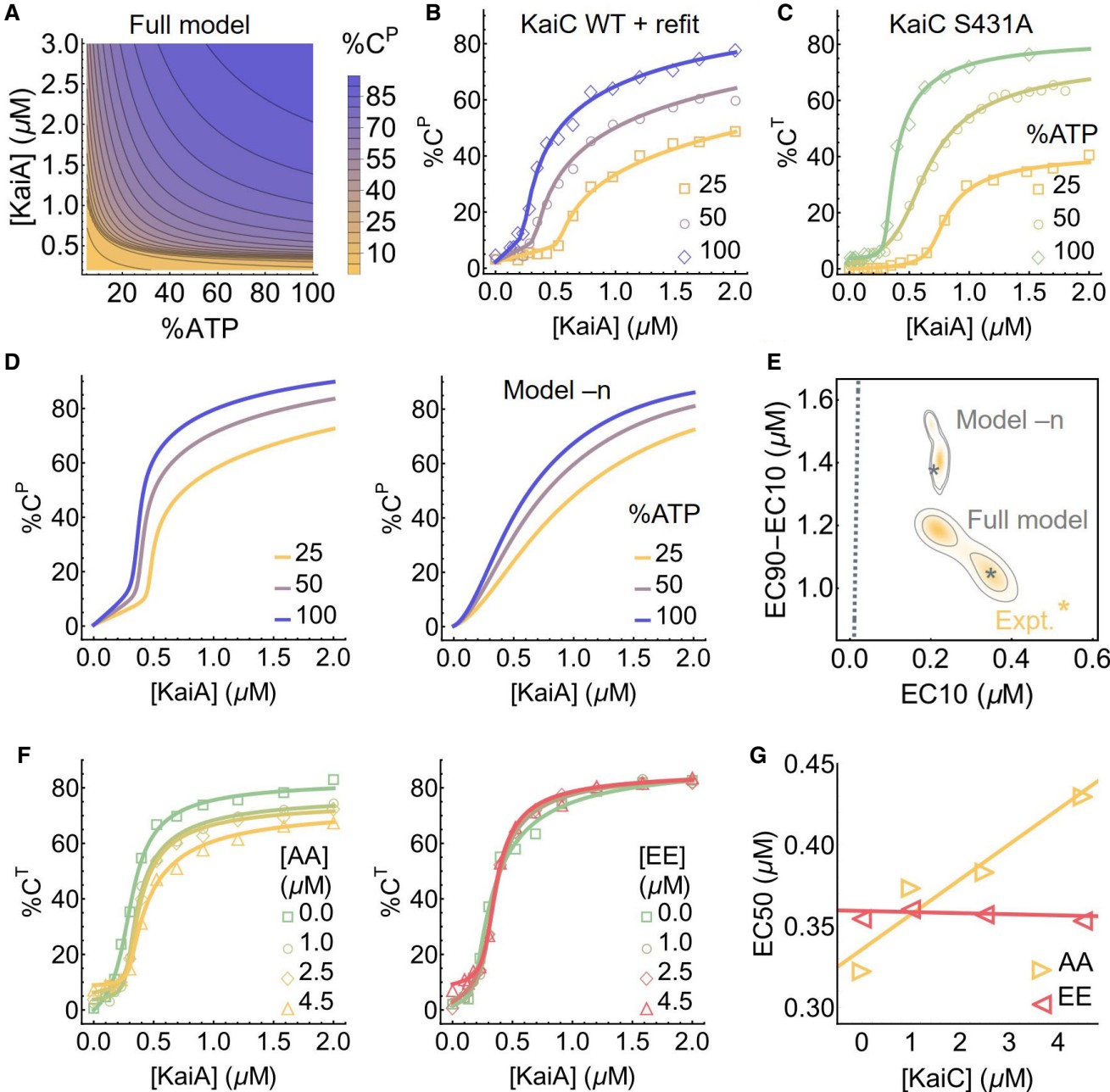

**Figure 3. Substrate competition explains KaiC phosphorylation ultrasensitivity.**

A   The predicted stimulus–response relation of the total steady-state KaiC phosphorylation level as a function of %ATP and [KaiA].

B   Experimentally determined stimulus–response function of KaiC at three %ATP conditions; the curves are based on refitting the best fit of the full model to the steady-state measurements.

C   Similar to B, but for KaiC S431A, which has only one phosphorylation site; the curves are based on independent fits to a simple phenomenological substrate competition model.

D   Cross sections of the stimulus–response relation at three %ATP, computed using the full model (left) and model –n (right).

E   Posterior distributions for the shape measures of the stimulus–response functions at 25% ATP predicted by the full model and model –n. The contours represent the 68% and 95% HDRs, and the gray stars represent the model best fits. The shapes of the stimulus–response functions are quantified using two metrics: EC10, which quantifies threshold-like behavior, and EC90–EC10, which quantifies switch-like behavior. The shape measures of the experimentally determined stimulus–response function at 25% ATP is shown as the yellow star. The dashed line represents (EC10, EC90–EC10) = ($K/9$, $80K/9$), which characterizes the shape of a hyperbolic stimulus–response function $[A]/(K + [A])$ that has no switching or thresholding.

F   The stimulus–response functions of KaiC S431A at 100% ATP in the presence of KaiC S431A/T432A (AA; left) and S431E/T432E (EE; right) phosphomimetic mutants to probe the effect of kinetic competition on KaiC phosphorylation.

G   The relations between EC50 (the midpoint of a stimulus–response function) and KaiC AA/EE concentrations quantified using the curves shown in (F).

Source data are available online for this figure.

## A substrate competition mechanism underlies ultrasensitivity in KaiC phosphorylation

What is the mechanism of ultrasensitivity in KaiC phosphorylation? Given that each KaiC subunit has two phosphorylation sites, a plausible explanation is multisite phosphorylation, whereby the concentration of the maximally phosphorylated state exhibits an ultrasensitive dependence on the kinase concentration (Gunawardena, 2005) (or in this case, the nucleotide exchange factor concentration), even if each consecutive phosphorylation step follows mass-action kinetics. To examine this possibility, we measured the stimulus–response relation of the KaiC S431A mutant, which has only one phosphorylation site, and the results show ultrasensitivity comparable to that of the WT protein (Fig 3C). Furthermore, because KaiC is its own phosphatase, it violates the assumption of distributivity (i.e., at most one modification takes place before the dissociation of the enzyme and substrate) (Gunawardena, 2005). Multisite phosphorylation thus cannot explain the observed ultrasensitivity.

In the ensemble of parameter sets, the KaiA dissociation constant of the ADP-bound (but not ATP-bound) U phosphoform ($C_{DP}^U$) is in or below the nanomolar range, much smaller than that of any other state of KaiC (Fig 1D). This is consistent with recent single molecule observations that the unphosphorylated form of KaiC can bind very tightly to KaiA (Mori *et al*, 2018), and native mass spectrometry measurements that KaiA binding to KaiC is enhanced by ATP hydrolysis, which would be needed to produce ADP-bound KaiC (Yunoki *et al*, 2019). Here, we argue that the key to understanding the origin of ultrasensitivity in the model lies in the differential binding affinity of KaiA to the ADP- and ATP-bound states of KaiC. We note here that since the model does not consider the hexameric structure of KaiC, we cannot rule out possible hexameric cooperative effects that may contribute to ultrasensitivity.

In the model, differential KaiA binding affinity leads to the following dynamics: KaiA promotes phosphorylation by catalyzing the exchange of the bound ADP for ATP, but this process is in a kinetic competition with ATP hydrolysis, which returns KaiC to the ADP-bound state. At the beginning of the phosphorylation reaction, almost all the KaiA is bound to $C_{DP}^U$ (Fig 2A and B) due to its high abundance and high affinity for KaiA (Fig 1D). When [KaiA] is low, the competition between nucleotide exchange and hydrolysis in the U phosphoform reaches a steady-state where [$C_{DP}^U$] stays above [KaiA] (Fig 2A, dashed curves). Therefore, KaiA stays trapped by $C_{DP}^U$ and the phosphorylation products (mostly T) cannot undergo nucleotide exchange. In the absence of KaiA, the autophosphatase activity of KaiC dominates, and the phosphorylation products revert back to the U phosphoform.

When [KaiA] is high, however, the competition between nucleotide exchange and hydrolysis in the U phosphoform pushes $C_{DP}^U$ below [KaiA] (Fig 2A, solid curves), which frees KaiA to catalyze the nucleotide exchange reactions of the phosphorylation products. Once the flux of phosphorylation, KaiA binding, and nucleotide exchange outweighs that of hydrolysis, dephosphorylation, and KaiA unbinding, the phosphorylation products stay phosphorylated at steady state. Furthermore, the formation of phosphorylated products positively feeds back to deplete $C_{DP}^U$, further removing a KaiC state that traps KaiA and leading to rapid saturation of

phosphorylation past the [KaiA] threshold. The [KaiA] threshold for phosphorylation depends on %ATP (Fig 3A), because when %ATP is low, more KaiA is needed to counteract the reduced ADP-to-ATP exchange rate.

This mechanism is a form of substrate competition (Buchler & Louis, 2008; Ferrell & Ha, 2014b), a previously identified general scheme where the kinetic competition of multiple substrates for enzyme binding leads to ultrasensitivity. Here, KaiA plays the role of the enzyme, while the ADP-bound U phosphoform and the T phosphoform (as well as the S and D phosphoforms to a lesser extent due to phosphorylation ordering) are the substrates that compete for KaiA binding. However, the fact that the phosphorylated and unphosphorylated forms of KaiC can interconvert through phosphotransfer reactions distinguishes the Kai system from a typical substrate competition scheme, where the substrates cannot interconvert.

The model suggests that the U phosphoform plays a special role in generating ultrasensitivity due to the significant difference in the affinity of KaiA for its ATP- vs. ADP-bound states (Fig 1D). This observation leads to two testable predictions. First, the amount of KaiA required to activate phosphorylation should be higher when more U phosphoform is present. We tested this prediction experimentally by measuring the stimulus–response relation of KaiC S431A in the presence of KaiC S431A/T432A (AA), which mimics the U phosphoform, or KaiC S431E/T432E (EE), which mimics the D phosphoform. The KaiC AA and EE mutants act as competitors for the KaiA—KaiC interaction (Fig 3F). Consistent with the hypothesis, the EC50 (i.e., the midpoint of the ultrasensitive switch) is positively correlated with the concentration of KaiC AA, while varying the concentration of KaiC EE has little effect (Fig 3G). Interestingly, the presence of KaiC AA in an oscillatory reaction also appears to reduce the amplitude of the oscillation, but the effect is most pronounced at low %ATP (Fig EV3C).

Second, the substrate competition mechanism suggests that the model should exhibit weaker nonlinearity if KaiA has the same affinity to ATP- vs. ADP-bound states of a given KaiC phosphoform. To computationally test this prediction, we constructed simplified models where KaiA on/off rates are set to be independent of the nucleotide-bound state (model –n) or phosphorylation state (model –p) and fit the new models to the experimental data *ab initio*. Consistent with the prediction, decoupling KaiA on/off rates from the nucleotide-bound states results in a significant loss of ultrasensitivity (Fig 3D right and Fig EV5A). Model –p by contrast behaves similarly to the full model (Fig EV5C); consistent with the substrate competition mechanism, the ADP-bound states of KaiC in model –p have higher affinity to KaiA than the ATP-bound states, regardless of the phosphorylation state (Fig EV5B). We quantify the effects of such model reductions by computing the Bayes factor, which is a metric for systematic model comparison that favors goodness of fit but penalizes model complexity and parameter fine tuning (MacKay & Kay, 2003); it is similar to the Bayesian information criterion (Schwarz, 1978), but makes no asymptotic assumptions. The analysis shows that the loss of ultrasensitivity in model –n degrades the fit quality significantly, while model –p is only marginally worse than the full model (Table 1). Interestingly, a model where the KaiA on/off rates are completely independent of the state of KaiC (model –n,–p; Fig EV5D and E) is much worse than either model –n or model –p (Table 1). We conclude that the nucleotide-bound state of

**Table 1. Effects of differential KaiA (un)binding kinetics.**

| Model | Log likelihood | | | Bayes factor[a] |
|---|---|---|---|---|
| | Phosphorylation | Dephosphorylation | Hydrolysis | |
| Full model | 422.9 | 346.8 | −0.8 | 1 |
| −n[b] | 249.2 | 275.4 | −0.3 | 10.4 |
| −p[c] | 392.4 | 303.2 | −2.4 | 2.2 |
| −n,−p | 204.4 | 266.4 | −2.6 | 19.7 |

[a]We define the Bayes factor as the ratio of the marginal likelihood function of the full model over that of the simplified models. We adopt the convention that a Bayes factor larger than 3.2 is substantial evidence against the model (Vyshemirsky & Girolami, 2008).
[b]−n: on/off rates decoupled from nucleotide-bound state.
[c]−p: on/off rates decoupled from phosphorylation state.

KaiC plays a key role in regulating its interaction with KaiA and thus in determining phosphorylation kinetics.

## Substrate competition may underlie metabolic compensation

Finally, we consider the implications of the ultrasensitivity for the full oscillator. For the sake of clarity, we make a distinction in this section among three subpopulations of KaiA: the sequestered KaiA, which refers to inactive KaiA in a KaiABC complex; the active KaiA, which refers to (free or bound) KaiA not sequestered by KaiB; and the free KaiA, which is not associated with either KaiB or KaiC.

We first consider the current understanding of how KaiA activity is regulated over the circadian cycle. It is well established that the regulation of KaiA during nighttime by KaiB plays an essential role in producing the negative feedback loop. At dusk, the buildup of KaiC D and S phosphoforms triggers the binding of KaiB to CI (Rust et al, 2007; Chang et al, 2012; Mutoh et al, 2013; Phong et al, 2013; Lin et al, 2014; Snijder et al, 2017; Tseng et al, 2017; Mukaiyama et al, 2018) and subsequently the sequestration of KaiA by CI-bound KaiB (Chang et al, 2012). In the absence of active KaiA, the CII domain autodephosphorylates, and the KaiABC ternary complex disassembles (Snijder et al, 2017) at dawn as KaiC reaches its unphosphorylated state (Tomita et al, 2005), freeing KaiA and readying the clock for the next cycle.

This understanding of the negative feedback loop implies that the sequestration of KaiA by KaiB is a source of nonlinearity in the system that is critical for oscillation. Indeed, in many models of the Kai oscillator, the complete sequestration of KaiA during dephosphorylation is either a built-in or required feature for stable oscillation (e.g., Yoda et al, 2007; van Zon et al, 2007; Phong et al, 2013; Paijmans et al, 2017b). However, our observation that phosphorylation is suppressed nonlinearly at low [KaiA] suggests that complete sequestration of KaiA by KaiB is not necessary to prevent phosphorylation at night. Indeed, there is mounting evidence that KaiB sequestration by itself is not entirely responsible for inactivating KaiA during dephosphorylation. Specifically, measurements using native mass spectrometry, co-immunoprecipitation (co-IP), and native PAGE suggest that there is a significant amount of $KaiA_2C_6$ complex (Kageyama et al, 2006; Brettschneider et al, 2010) and free KaiA (Qin et al, 2010a) throughout the entire phosphorylation cycle.

To confirm that KaiA is not fully sequestered by KaiBC complexes, we used immunoprecipitation of FLAG-tagged KaiB to monitor the amount of uncomplexed KaiA in supernatant, which we interpret to be a measure of active KaiA concentration (Figs 4A and

EV3D). The experiment shows that there is indeed a substantial amount of active KaiA in solution in the first half of the dephosphorylation stage of the oscillation. Taken together, these results suggest that either the binding of KaiA to KaiBC has lower affinity than previously assumed, or that the sequestration kinetics are slow compared to the length of the dephosphorylation stage. In either case, some KaiA appears to remain free of KaiABC complexes at all times during the oscillation.

Given these results, we consider the role ultrasensitivity may play in regulating KaiA's ability to stimulate KaiC phosphorylation at nighttime. In particular, the fact that the phosphorylation threshold scales with %ATP suggests that ultrasensitivity may also lead to insensitivity of the period of the Kai oscillator to %ATP (Phong et al, 2013), a phenomenon termed "metabolic compensation" (Johnson & Egli, 2014). As a proof of principle, we examine this possibility using a simple model of the Kai oscillator proposed by Phong et al (2013), which we refer to as the Phong model. The Phong model explicitly keeps track of the monomer phosphorylation cycle and uses KaiB binding to the S phosphoform to generate negative feedback (Fig 4B). In the Phong model, the KaiA sequestration affinity is effectively infinite. In light of the co-IP experiment, we modify the model by assuming that the KaiA sequestration reaction is in a quasi-equilibrium with a dissociation constant for KaiA binding to the KaiBC complex, $K_D$ (Fig 4C; see Appendix for mathematical details). When $K_D$ is small (i.e., $< 10^{-3}$ μM), the modified model exhibits the same robust oscillations as the original model over a large range of %ATP, but the range of %ATP that allows for stable oscillation shrinks as $K_D$ increases (Fig 4E top), and the model is unstable when $K_D$ is in the micromolar range regardless of %ATP.

In the original Phong model, the dependence of KaiC phosphorylation on KaiA is described by a Michaelis–Menten-like function with no ultrasensitivity. In this scenario, a small increase in active KaiA leads to a proportional increase in phosphorylation, making the dephosphorylation phase of the clock strongly dependent on the strength of KaiB-mediated KaiA sequestration. To test whether ultrasensitivity can increase the robustness of oscillations in the model, we introduce a phenomenological patch to the model in the form of an ultrasensitive KaiA threshold to the phosphorylation rate function, which varies as a function of %ATP and U phosphoform concentration (Fig 4D; see Appendix for mathematical details). Given that the ultrasensitivity is a result of substrate competition, this modification effectively introduces an inhibitory interaction between the U phosphoform and KaiA (Fig 4B, dashed arrow). This modification amounts to the assumption that the EC50 measured at

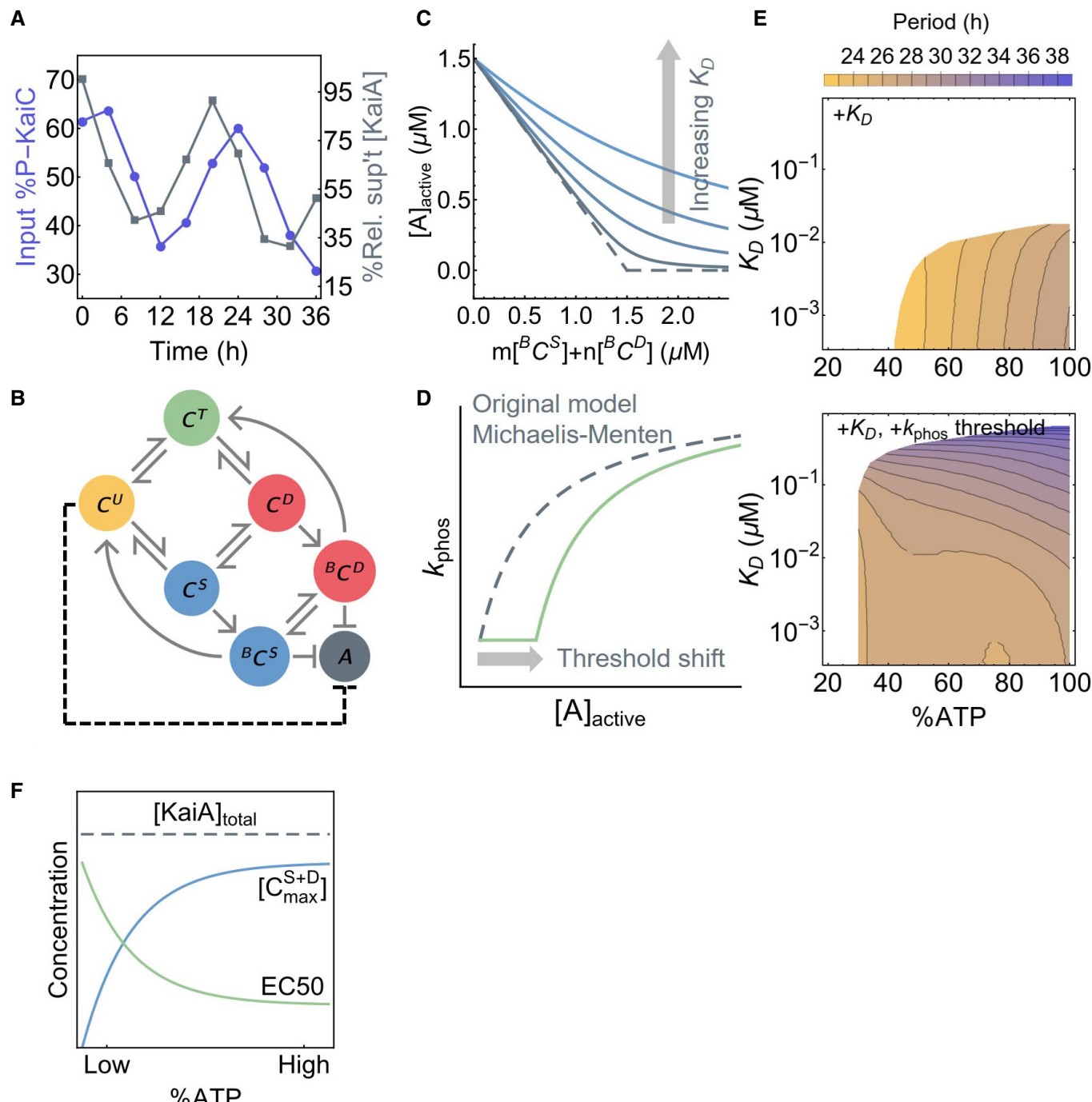

**Figure 4. Ultrasensitivity contributes to metabolic compensation.**

A   The time series of the total input KaiC phosphorylation level (purple, left scale) and residual KaiA concentration not precipitated with KaiB-FLAG (gray, right scale).

B   A schematic of the oscillator model by Phong *et al* (2013). Here, $^{B}C^{S}$ and $^{B}C^{D}$ represent the KaiB-bound S and D phosphoforms, respectively, which can sequester KaiA. The dashed line represents the effect of introducing ultrasensitivity to the model.

C   A cartoon representation of introducing a KaiA sequestration affinity, $K_D$, into the Phong model. The original model has an effectively infinite sequestration affinity (dashed curve).

D   A cartoon representation of introducing a KaiA threshold to the Michaelis–Menten-type phosphorylation rate constant in the Phong model.

E   The period of the oscillator model as a function of %ATP and $K_D$, a measure of KaiA sequestration affinity, without (top) or with (bottom) a phosphorylation threshold. All model simulations were done with 3.5 μM of KaiC and 1.5 μM of KaiA. White regions indicate unstable or no oscillation.

F   The extent to which KaiA can be sequestered by KaiB depends on the maximal S and D phosphoform concentration, $[C^{S+D}_{max}]$, achieved over the phosphorylation cycle. The scaling of the EC50 of the phosphorylation stimulus–response function, which is a measure of the capacity of the U phosphoform to suppress KaiA activity, compensates for the scaling of $[C^{S+D}_{max}]$ with %ATP.

steady state (Fig 3A) in the absence of KaiB corresponds to the active KaiA concentration required to re-enter the phosphorylation phase at the trough of the circadian oscillation. Remarkably, the resulting model can generate stable oscillations over a larger range of both %ATP and $K_D$ conditions, and the period of oscillation is made much less dependent on %ATP (Fig 4E bottom). This observation suggests that ultrasensitivity in KaiC phosphorylation plays a role in clock stability that complements the function of KaiB-dependent KaiA sequestration.

Why does ultrasensitivity in KaiC phosphorylation allow for metabolic compensation? The binding of KaiB to KaiC, and thus the sequestration and inactivation of KaiA, depends on S431 phosphorylation of KaiC (i.e., the S and D phosphoforms). At low %ATP, the maximal S and D concentrations, $[C_{max}^{S+D}]$, are lower (Fig 4F). Thus, the maximal amount of KaiA sequestered by the KaiBC complex is smaller. This is problematic for the stability of the clock at low % ATP, since the active KaiA can promote premature KaiC U → T phosphorylation of some molecules, which can lead to phase decoherence that manifests as decay of the oscillation (Fig EV3F). The ultrasensitive stimulus–response that we report here implies that a finite amount of KaiA must be liberated from KaiB before there is a noticeable impact on KaiC phosphorylation. In other words, the inhibitory effect of ultrasensitivity is a synchronization mechanism at night. Importantly, the EC50 of the stimulus–response function scales with %ATP, such that the capacity of phosphorylation suppression by $C^U$ is enhanced at low %ATP, which compensates for weaker KaiB-mediated KaiA sequestration (Fig 4F). The compensatory relation between these two mechanisms as %ATP varies is likely key to period stability. This relation likely also contributes to the scaling of the phosphorylation limit cycle size with %ATP. At higher %ATP, the EC50 is smaller, and thus, more KaiA needs to be sequestered to trigger dephosphorylation, which implies that higher concentrations of the S and D phosphoforms need to accumulate to enable KaiB binding. Since KaiC phosphorylation is ordered, this means that the T phosphoform concentration scales with %ATP as well.

# Discussion

In this work, we undertook a data-driven kinetic modeling approach to understand the mechanism and metabolic sensitivity of the KaiA—KaiC subsystem, part of the *S. elongatus* circadian oscillator. We constructed a detailed yet mechanistically naive kinetic model, which was fit to extensive experimental measurements of KaiC phosphorylation kinetics within a Bayesian parameter estimation framework. Approaches that are similar in spirit have been pursued in eukaryotic systems (e.g., Forger & Peskin, 2003; Kim & Forger, 2012; Locke *et al*, 2005; Mirsky *et al*, 2009; Relógio *et al*, 2011). However, owing to the greater complexity of eukaryotic clocks, these studies combined direct experimental measurements, cost function optimization, and hand tuning of selected parameters to account for unknown or unconstrained biochemical processes. Because the Kai system can be studied as a well-defined mixture of purified components, the participating molecular species are known, and all the parameters in the model can be treated in a consistent manner to enable objective comparison of mechanisms underlying collective oscillations.

This data-driven approach is to be contrasted with the more common hypothesis-driven approach, whereby a model is built to examine how features of the oscillator arise from proposed mechanisms. This hypothesis-driven approach has been employed extensively in the study of the cyanobacterial clock. These studies have revealed insights into specific aspects of the oscillator function, such as entrainment (Brettschneider *et al*, 2010; Leypunskiy *et al*, 2017), synchronization (Sasai, 2019; Yoda *et al*, 2007; van Zon *et al*, 2007), irreversibility (Cao *et al*, 2015), and robustness against variations in temperature (François *et al*, 2012; Hatakeyama & Kaneko, 2012; Kidd *et al*, 2015; Murayama *et al*, 2017), ATP/ADP concentration (Paijmans *et al*, 2017a; Phong *et al*, 2013), protein copy numbers (Brettschneider *et al*, 2010; Chew *et al*, 2018; Lin *et al*, 2014), and environmental noise in general (del Junco & Vaikuntanathan, 2020; Monti *et al*, 2018; Pittayakanchit *et al*, 2018). This hypothesis-driven approach is pedagogically powerful but relies on the ingenuity of the modeler to identify *a priori* which mechanisms should be included. These approaches give little indication of the range of the parameter space consistent with a proposed mechanism, which makes it difficult to quantify the uncertainties of model predictions and validate them experimentally.

Not all parameters in our model were fully constrained by the data, as expected given the complexity of the model (Gutenkunst *et al*, 2007). Nevertheless, the ensemble of parameter sets still led to consistent predictions. In particular, the model revealed unexpected ultrasensitivity in KaiC phosphorylation as a function of KaiA, which we confirmed experimentally. The source of ultrasensitivity in the model is a substrate competition mechanism that arises from the differential affinity of ADP- and ATP-bound KaiC for KaiA. Previous studies have considered the importance of the differential affinity of KaiA for KaiC states but have focused on phosphorylation (Mori *et al*, 2018; Paijmans *et al*, 2017b). We note here that the ultrasensitivity in KaiC phosphorylation that we discovered and the ultrasensitivity in KaiB-dependent KaiA sequestration that arises from opposing S and T phosphorylations within hexamers (Lin *et al*, 2014) perform fundamentally different roles. Specifically, the cooperative binding of KaiB to KaiC during the dephosphorylation stage provides the nonlinear, delayed negative feedback loop essential for oscillation (Novák & Tyson, 2008), whereas the mutual inhibition between KaiA and ADP-bound KaiC, which effectively provides positive feedback on KaiA's activity, generates a KaiC phosphorylation threshold that contributes to period stability as %ATP changes. The presence of nonlinearities and delayed feedback at multiple steps in a molecular oscillator allows the system to achieve greater robustness (Dovzhenok *et al*, 2015; Jolley *et al*, 2012; Kim & Forger, 2012; Pett *et al*, 2016).

We hypothesized that ultrasensitivity in KaiC phosphorylation plays a role in stabilizing the oscillator at low %ATP conditions by suppressing premature phosphorylation during the dephosphorylation stage and thus promoting phase coherence. Currently, the Kai oscillator model most robust against yet tunable by metabolic conditions appears to be that of Paijmans *et al* (2017a,b). In the Paijmans model, metabolic compensation is achieved both at the hexamer and ensemble level. At the hexamer level, the onset of dephosphorylation is primarily controlled by the antagonistic effects of the T and S phosphoforms. Since fewer subunits in the T phosphoform accumulate at low %ATP, fewer subunits in the S phosphoform are needed to trigger dephosphorylation; therefore, the reduced

amplitude of oscillation counteracts the slower phosphorylation rate at low %ATP. At the ensemble level, low %ATP limits the fraction of hexamers that are able to trigger dephosphorylation before the onset of KaiB-mediated delayed inhibition; this makes the dephosphorylation phase shorter, which compensates for the longer phosphorylation phase. It is worth noting that, unlike the data, the Paijmans model is not oscillatory when %ATP reaches below 50%, partly due to phase decoherence during dephosphorylation, an issue that can potentially be addressed with ultrasensitivity in KaiC phosphorylation. In our model, the coupling between KaiA binding affinity and KaiC nucleotide-bound states is critical in generating ultrasensitivity, a feature that is missing in the Paijmans model. It remains an open question whether a hexameric model with no such coupling can nevertheless produce ultrasensitivity in KaiC phosphorylation (see Appendix for further comparison between this work and the Paijmans model). The ultimate goal of this line of research is to obtain a molecularly detailed model of the complete Kai oscillator that is constrained by direct fitting to empirical measurements.

In *S. elongatus*, the Kai oscillator is embedded in a transcription–translation feedback loop (Kitayama *et al*, 2008; Qin *et al*, 2010b; Zwicker *et al*, 2010). However, with the exception of peroxiredoxin oxidation cycles (Edgar *et al*, 2012; O'Neill *et al*, 2011), cell-autonomous circadian rhythms in most eukaryotes are thought to be generated by interlocked transcription-translation feedback loops (Novák & Tyson, 2008); the cooperative autoregulation of transcription is a key source of nonlinearity and robustness in the circuit (e.g., Brown *et al*, 2012; Gonze *et al*, 2002; Leloup & Goldbeter, 2003; Leloup *et al*, 1999; Locke *et al*, 2005). Our results raise the possibility that post-translational modifications and protein–protein interactions may also contribute to robustness generally in circadian clocks by introducing ultrasensitivity, even if these processes do not generate self-sustaining rhythms that can be decoupled from transcription. Overall, it is clear that post-translational steps such as (de)phosphorylation (Fustin *et al*, 2018; Gallego & Virshup, 2007; Reischl & Kramer, 2011; Zhou *et al*, 2015), protein degradation (Gallego & Virshup, 2007; Reischl *et al*, 2007), and complex formation (Kim & Forger, 2012) play an important role in eukaryotic circadian oscillators, but to our knowledge there is currently no complete experimental characterization of the stimulus–response relations of these processes.

# Materials and Methods

## Computational methods

### Treatment of nucleotide exchange

Here, we derive (4) in Results. The nucleotide exchange process can be modeled as a two-step reaction that includes an apo intermediate state of KaiC, i.e.,

$$^A C_{TP} \underset{k_{on}^{TP}}{\overset{k_r^{TP}}{\rightleftharpoons}} {}^A C \underset{k_r^{DP}}{\overset{k_{on}^{DP}}{\rightleftharpoons}} {}^A C_{DP} \tag{6}$$

where we have omitted the free ATP and ADP from the chemical equation. Here, $k_r^{TP}$ and $k_r^{DP}$ are the dissociation rate constants and $k_{on}^{TP}$ and $k_{on}^{DP}$ are the binding rate constants for ATP and ADP, respectively. Since KaiC requires nucleotides for hexamerization

(Hayashi *et al*, 2003, 2006; Mutoh *et al*, 2013), the apo state of KaiC is presumably both thermodynamically and kinetically unstable in the presence of a saturating amount of nucleotide (5 mM in our experiments). Therefore, under the assumption that the KaiC apo state is in a quasi-steady state throughout the reactions, we can eliminate the apo state and model nucleotide exchange as a one-step reaction

$$^A C_{TP} \underset{k_{TP}^A}{\overset{k_{DP}^A}{\rightleftharpoons}} {}^A C_{DP} \tag{7}$$

where

$$k_{TP}^A = k_r^{DP} \frac{[ATP]}{[ATP] + K_{on}[ADP]} \tag{8}$$

$$k_{DP}^A = k_r^{TP} \left( 1 - \frac{[ATP]}{[ATP] + K_{on}[ADP]} \right) \tag{9}$$

and $K_{on} = k_{on}^{DP}/k_{on}^{TP}$ is a ratio of the two nucleotide binding rate constants.

We make two further simplifying assumptions. First, we assume the on rates are completely diffusion controlled and are thus the same for ATP and ADP, which allows us to set $K_{on} = 1$. Second, based on fit results (Fig EV2F) showing that the posterior for $k_r^{TP}$ has a long tail to negative infinity in log space, we follow the approach proposed by Transtrum and Qiu (2014) and set $k_r^{TP} = 0$; i.e., the dwell time of ATP-bound states is sufficiently long that a bound ATP cannot be released without first giving up its γ-phosphate group. This assumption implies that the only ways for KaiC to enter an ADP-bound state are through hydrolysis and phosphorylation, and solution ADP has no effect on the system except to slow down the ADP to ATP exchange process. With these two assumptions, we eliminate (9), and (8) reduces to (4).

### Model parameterization

In Results, we introduced a model parameterization scheme in which rate constants for phosphotransfer, nucleotide exchange, KaiA (un) binding, and ATP hydrolysis reactions depend on the participating molecular species. Although we use this independent rate scheme to interpret the model, including computing the sensitivity ODEs, during the fitting itself we represent state-dependent effects by modifying each of the six basic rate constants ($k_p$, $k_d$, $k_a$, $k_b$, $k_r^{DP}$, and $k_h$) by multiplicative $\Delta k$ factors. For example, the KaiA dissociation rate $k_b^{T,TP} = k_b \Delta k_b^{T,TP}$ is represented by the product between a base rate $k_b$ and a modifier $\Delta k_b^{T,TP}$ (compare Fig EV1 and Appendix Fig S5). The multiplicative-factor scheme introduces 38 $\Delta k$ parameters. Because of the requirement for detailed balance (see below), only 34 of these parameters are free; these free parameters are listed on Table 2. The advantage of the multiplicative parameterization scheme is that it facilitates $\ell^1$ regularization, discussed below.

### Detailed balance

All elementary reactions, except ATP hydrolysis and nucleotide exchange, are assumed to be able to reach equilibrium, and thus, the free energy change over each reversible cycle must be equal to zero. In practice, this means that the product of all rate constants in

Table 2. Full model parameters and their priors.

| Category | State-dependent effect | Parameters | Prior | Unit |
|---|---|---|---|---|
| Basic | N/A | $k_h, k_p, k_d, k_a, k_b, k_r^{DP}$ | $10^{\mathcal{N}(\mu,\,3)}$[b] | $s^{-1}$[d] |
| Nucleotide exchange | KaiA & phos.[a] | $\Delta k_{TP}^{A,T}, \Delta k_{TP}^{A,S}, \Delta k_{TP}^{A,D}$ | $10^{Laplace\,(\mu,1)}$[b] | N/A |
| Hydrolysis | phos. | $\Delta k_h^T, \Delta k_h^S, \Delta k_h^D$ | | |
| | KaiA & phos. | $\Delta k_h^{A,U}, \Delta k_h^{A,T}, \Delta k_h^{A,S}, \Delta k_h^{A,D}$ | | |
| KaiA on | nuc.[a] & phos. | $\Delta k_a^{U,DP}, \Delta k_a^{D,DP}, \Delta k_a^{D,TP}$ | | |
| KaiA off | nuc. & phos. | $\Delta k_b^{U,DP}, \Delta k_b^{T,DP}, \Delta k_b^{S,DP},$ $\Delta k_b^{D,DP}, \Delta k_b^{T,TP}, \Delta k_b^{S,TP},$ $\Delta k_b^{D,TP}$ | | |
| (De)phosphorylation | phos. | $\Delta k_p^{US}, \Delta k_d^{SU}, \Delta k_p^{TD}, \Delta k_d^{DT},$ $\Delta k_p^{SD}, \Delta k_d^{DS}$ | | |
| | KaiA & phos. | $\Delta k_p^{A,UT}, \Delta k_d^{A,TU}, \Delta k_p^{A,TD},$ $\Delta k_d^{A,DT}, \Delta k_p^{A,SD}, \Delta k_d^{A,DS},$ $\Delta k_p^{A,US}, \Delta k_d^{A,SU}$ | | |
| Global error | N/A | $\sigma^2$ | Inv-Gamma (1, 0.01) | $\mu M^2$ |
| Initial conditions | N/A | $[C_{TP}^U]_0, [C_{DP}^U]_0, [C_{TP}^T]_0, [C_{DP}^T]_0$ $[C_{TP}^S]_0, [C_{DP}^S]_0, [C_{TP}^D]_0, [C_{DP}^D]_0$ | Dirichlet (a)[c] | $\mu M$ |

[a]phos., phosphoform; nuc., nucleotide-bound state.
[b]The mean of the priors, $\mu$, is zero unless specified by Table 5.
[c]$a = (20,100,1,1,1,1,1,1)$; points drawn from the distribution are scaled by the total KaiC concentration. The support of the Dirichlet distribution implies that only seven of the eight initial conditions are free fitting parameters.
[d]or $s^{-1} \cdot \mu M^{-1}$ for the second-order rate constant $k_a$.

Table 3. Detailed balance conditions.

| Cycle | Detailed balance condition |
|---|---|
| $\{C_{TP}^S, C_{DP}^D, {}^A C_{DP}^D, {}^A C_{TP}^S\}$ | $\delta k_a^{S,TP} = \Delta k_b^{S,TP} \dfrac{\Delta k_d^{A,DS}}{\Delta k_d^{A,SD}} \dfrac{\Delta k_b^{D,DP}}{\Delta k_b^{D,DP}}$ |
| $\{C_{TP}^T, C_{DP}^D, {}^A C_{DP}^D, {}^A C_{TP}^T\}$ | $\delta k_a^{T,TP} = \Delta k_b^{T,TP} \dfrac{\Delta k_d^{A,DT}}{\Delta k_d^{A,TD}} \dfrac{\Delta k_b^{D,DP}}{\Delta k_b^{D,DP}}$ |
| $\{C_{TP}^U, C_{DP}^T, {}^A C_{DP}^T, {}^A C_{TP}^U\}$ | $\delta k_a^{T,DP} = \Delta k_b^{T,DP} \dfrac{\Delta k_p^{A,UT}}{\Delta k_d^{A,TU}}$ |
| $\{C_{TP}^U, C_{DP}^S, {}^A C_{DP}^S, {}^A C_{TP}^U\}$ | $\delta k_a^{S,DP} = \Delta k_b^{S,DP} \dfrac{\Delta k_p^{A,US}}{\Delta k_d^{A,SU}}$ |

the forward direction of each cycle listed in Table 3 must be equal to that in the reverse direction (see also Appendix Fig S5A). This introduces an additional algebraic constraint for each such cycle, which is used to eliminate one free $\Delta k$ parameter. In total, one can eliminate four parameters.

### Fitting data set

To constrain the model parameters, we collected experimental measurements that characterized different aspects of the KaiA—KaiC subsystem, which are summarized in Table 4.

The dephosphorylation reaction data taken from Rust et al (2007) constrain the dephosphorylation rates and the ATP hydrolysis rates of KaiC in the absence of KaiA, because the model structure dictates that dephosphorylation requires alternating phosphotransfer and hydrolysis reactions. The maximum ADP production rate of KaiC in the presence of 1.2 µM of KaiA was reported to be 29.8 ± 5.1 KaiC$^{-1}$ per day (Terauchi et al, 2007), which we take as an upper bound on the average CII hydrolysis rate in phosphorylation reactions with [KaiA] = 0.375, 0.75, and 1.50 µM.

Because the phosphorylation reactions were measured in the presence of varying %ATP and higher [ADP] inhibits

phosphorylation by slowing down nucleotide exchange (see equation 4), such measurements provide indirect constraints on the nucleotide exchange rate. Similarly, because the reactions were measured in the presence of varying [KaiA] conditions, they directly constrain the phosphorylation rates of KaiC with and without KaiA, as well as the KaiA binding affinity, i.e., the ratio of KaiA on/off rates. Although there are direct experimental measurements of KaiA binding and dissociation (Kageyama et al, 2006; Mori et al, 2018), these results cannot be directly mapped onto model rate constants. This is primarily because the KaiC nucleotide-bound fractions are not measured in these experiments, or, in the case of phosphomimetic mutants, it is unclear if the mutations affect nucleotide binding affinities. As a consequence, the experimental constraints on KaiA on/off rates enter through the priors rather than the likelihood function, in contrast to the other data (see below).

### Initial conditions

For each phosphorylation reaction in Table 4, we solve the ODE model with the corresponding [KaiA] and %ATP condition. The predicted phosphoform composition, as well as the ATP hydrolysis rate when appropriate, is compared to the experimental measurements in the Bayesian parameter estimation formalism described below. However, since the experimental data do not resolve the initial conditions for all 16 KaiC states in the model, we have chosen to directly estimate the initial concentrations by treating them as free parameters in the fitting procedure. We take $t = 0$ to be the time point at which KaiA is mixed with KaiC, and thus, all eight KaiA-bound KaiC states have zero concentration at the onset of the experiment. Because total KaiC concentration is conserved, this introduces seven additional parameters (see Table 2).

**Table 4.  Fitting dataset.**

| Measurement (source) | Temperature (°C) | [KaiA] (μM) [a] | ATP | Time points | Phosphoform |
|---|---|---|---|---|---|
| Phosphorylation (this work) | 30 | 0.375, 0.75, 1.50, 3.00, 4.50, 6.00 | 10, 25, 100%[b] | 8 | U, T, D [c] |
| Dephosphorylation (Rust *et al*, 2011) | 30 | 1.4 | 5 mM | 21 | |
| ADP production (Terauchi *et al*, 2007) | 30 | 1.2 | 1 mM | 1 | N/A |
| KaiA on/off rates (Kageyama *et al*, 2006) | 25 | Variable | 1 mM | N/A | Likely U |
| KaiA dwell time (Mori *et al*, 2018) | 25–28 | 1.0 | N/A | N/A | T, S, D [d] |

[a]We report here on the KaiA monomer concentration. However, since KaiA functions as a dimer, all KaiA concentration is divided by two in the models.
[b]%ATP, defined as 100%[ATP]/([ATP] + [ADP]); total [ATP] + [ADP] is held constant at 5 mM.
[c]The conservation of mass constraint implies that one of the four phosphoforms is not a free state variable. We have chosen the S phosphoform to be the constrained state variable.
[d]Phosphomimetic mutants.

For the dephosphorylation reaction, we do not estimate the initial conditions. To mimic the way the experiment was done, the dephosphorylation reaction is simulated in two stages. In the first stage, we assume that 3.4 μM of dephosphorylated KaiC is phosphorylated in the presence of 1.3 μM KaiA and 100% ATP for 20 h. Since the protein is initially dephosphorylated, we assume that $[C_{TP}^{U} = 3.4\mu M]$, while the concentrations of all other KaiC states are set to zero. In the second stage, we simulate the autodephosphorylation reaction after KaiA pull-down, which corresponds to eliminating all free KaiA as well as KaiA-bound KaiC states from the simulation. The amount of KaiC lost in the pull-down experiment was not reported in the original experiment (Rust *et al*, 2011). We therefore make the assumption that the amount of KaiC lost in the pull-down experiment in the simulation is exactly equal to that in the experiment for the purpose of computing the likelihood function.

**Bayesian parameter estimation**

We directly fit numerically integrated ODEs to experimental data in the Bayesian parameter estimation framework (Wasserman, 2000). The best fit model parameters, $\hat{\theta}$, are obtained from the maximum *a posteriori* estimator:

$$\hat{\theta} = \arg \max_{\theta} p(\theta|\mathcal{D}), \tag{10}$$

where $p(\theta|\mathcal{D})$ is the posterior distribution of the parameters $\theta$, conditioned on the training data set $\mathcal{D}$. Using Bayes' theorem, the posterior distribution can be written as

$$p(\theta|\mathcal{D}) = \frac{\mathcal{L}(\mathcal{D}|\theta)p(\theta)}{p(\mathcal{D})}. \tag{11}$$

Here, $p(\theta)$ is the prior distribution, which represents subjective belief in the model parameters $\theta$ prior to experimental input; $\mathcal{L}(\mathcal{D}|\theta)$ is the likelihood function, which represents a probabilistic model of the experimental data set $\mathcal{D}$ given a particular model $\mathcal{M}$ (implicit in the formulas) that depends on the parameters $\theta$; $p(\mathcal{D})$ is the evidence, which is analogous to the partition function in statistical mechanics. Note that the evidence $p(\mathcal{D})$ does not depend on the parameter choice and is thus an irrelevant constant for the purpose of parameter estimation. The specific choices for the functional forms of the likelihood function and priors are discussed further below.

**Model priors**

The priors for all model parameters used in Bayesian parameter estimation are given in Table 2. Here, the choice of the prior distributions is primarily motivated by the need for regularization (see below). In addition, as discussed above, the experimental measurements on KaiA binding kinetics are incorporated into the priors rather than the likelihood function (Table 5). Note that all the rate constants and their multiplicative factors are estimated in the log space (base 10). This ensures that all rate constants are positive.

**$\ell^1$ regularization**

As model complexity grows, the parameters become less constrained by the data. To address this problem, we impose sparsity on the state-dependent effects (i.e., the $\Delta k$ factors) through $\ell^1$ regularization (Tibshirani, 1996). This is accomplished in the Bayesian parameter estimation framework by using a Laplace prior centered at zero in the log parameter space (or one in the real space). Intuitively, the Laplace prior imposes sparsity by forcing the (marginalized) posterior distribution for each log $\Delta k$ to peak at zero unless there is experimental evidence in the fitting data set to suggest otherwise. Since the $\Delta k$s are multiplicative factors modifying the six basic rate constants, log $\Delta k = 0$ implies that the state-dependent rate is identical to that of the base rate. This method is directly analogous to the use of the lasso estimator in the context of linear least-squares models (Tibshirani, 1996). To see this, consider the Laplace distribution

$$p(\theta; b) = \frac{1}{2b} e^{-||\theta||_1/b} \tag{12}$$

where $||\theta||_1$ is the $\ell^1$-norm of $\theta$. Then from (11) the negative log-posterior distribution becomes

$$-\ln p(\theta|\mathcal{D}) = -\ln \mathcal{L}(\mathcal{D}|\theta) + \lambda||\theta||_1 + \text{constant} \tag{13}$$

where $\lambda = 1/b$. In a linear model $Y = X\beta + \epsilon$ where $Y$ is the response vector, $X$ is the design matrix, $\beta$ is the parameter vector, and $\epsilon$ is the error vector, the negative log-likelihood function reduces to the sum of squares $||Y - X\beta||_2^2/N$, where $N$ is the number of dependent variables. Thus, maximizing the posterior is equivalent to minimizing the sum of squares with an $\ell^1$ penalty, which is the lasso estimator.

**Table 5. Priors incorporating KaiA on/off constraints.**

| Parameter | Prior mean (μ) | Experimental measurements | Source |
|---|---|---|---|
| $k_a$ | $\log k_{a,exp}$ | $k_{a,exp} = 0.0279 \text{ s}^{-1}\cdot\mu M^{-1}$ | Kageyama et al (2006) |
| $k_b$ | $\log k_{b,exp}$ | $k_{b,exp} = 0.0663 \text{ s}^{-1}$ | |
| $\Delta k_b^{T,DP}, \Delta k_b^{T,TP}$ | $-\log \tau_{b,exp}^{T} k_{b,exp}$ | $\tau_{b,exp}^{T} = 1.0 \text{ s}$ | Mori et al (2018) |
| $\Delta k_b^{S,DP}, \Delta k_b^{S,TP}$ | $-\log \tau_{b,exp}^{S} k_{b,exp}$ | $\tau_{b,exp}^{S} = 0.43 \text{ s}$ | |
| $\Delta k_b^{D,DP}, \Delta k_b^{D,TP}$ | $-\log \tau_{b,exp}^{D} k_{b,exp}$ | $\tau_{b,exp}^{D} = 0.26 \text{ s}$ | |

### Likelihood function

To determine the functional form of the likelihood function, we consider a kinetic experiment where measurements on some observables $y$ are made at a set of time points $\{t_i\}$ with uncertainties $\{\sigma_i\}$. If we assume that the experimental errors $\sigma_i$ are independent and normally distributed, then the likelihood function is given by

$$\mathcal{L}(\mathcal{D}|\theta) = \prod_i \frac{1}{\sqrt{2\pi}\sigma_i} e^{-[y_{exp}(t_i)-y_{model}(t_i;\theta)]^2/2\sigma_i^2}. \tag{14}$$

In other words, the likelihood function gives the probability for observing a given data set provided that the model prediction is true. In practice, all posterior evaluations are done in the log space (base $e$) for numerical stability. Thus, taken together, (11) can be rewritten as,

$$\ln p(\theta|\mathcal{D}) = -\sum_i \frac{[y_{exp}(t_i) - y_{model}(t_i;\theta)]^2}{2\sigma_i^2} - \sum_i \ln \sqrt{2\pi}\sigma_i \\ + \ln p(\theta) + \text{constant}. \tag{15}$$

For the sake of simplicity, we assume that there is a single global error, $\sigma$, for all (de)phosphorylation measurements, which is then estimated during fitting as a hyperparameter (see Table 2).

The choice of the Gaussian likelihood function applies to all (de) phosphorylation data sets, but not the hydrolysis constraint, which only provides an upper bound on the average hydrolysis rate per day (Terauchi et al, 2007). Therefore, for the hydrolysis data a "half harmonic" is used as the log-likelihood function:

$$\ln \mathcal{L}(\mathcal{D}|\theta) = \begin{cases} -\frac{\left([ADP]_{model}(\theta)-[ADP]_{exp}\right)^2}{2\sigma_h^2}, & [ADP]_{exp} < [ADP]_{model} \\ 0, & 0 \leq [ADP]_{model} < [ADP]_{exp} \end{cases}. \tag{16}$$

The total amount of ADP produced by the model during a phosphorylation reaction over 12 h, $[ADP]_{model}$, is given by the sum of all $P_i$ production over time plus all ADP-bound KaiC states at $t = 12$ h.

The log-likelihood values from the appropriate phosphorylation, dephosphorylation, and hydrolysis reactions are added together to determine the log-likelihood of the data set for each given model parameter choice. Since the phosphorylation data set is much larger than the dephosphorylation data set, the fitting procedure tends to favor fitting the phosphorylation data set at the expense of fitting the dephosphorylation data set. To overcome this problem, the log-likelihood function for the dephosphorylation reaction is multiplied by a factor of 4 to increase the weight of the dephosphorylation data points.

### Model fitting procedure

To determine the posterior mode and the uncertainties associated with the estimate, we employ a heuristic combination of ensemble MCMC sampling and numerical optimization methods (Appendix Fig S2A). This fitting procedure can be divided into four steps that are analogous to those in a genetic algorithm:

1.  Initialization. An ensemble MCMC method evolves a set of random walkers (i.e., parameter sets) simultaneously; we thus begin by drawing 224 walkers from the prior distribution $p(\theta)$ and use these walkers for simulated annealing (Kirkpatrick, 1984; Kirkpatrick et al, 1983). In annealing, instead of sampling $p(\theta|\mathcal{D}) \propto \mathcal{L}(\mathcal{D}|\theta)p(\theta)$ (in the log space), a flattened distribution $\mathcal{L}(\mathcal{D}|\theta)^\beta p(\theta)$ is sampled with an annealing schedule of $\beta$ = 0.3, 0.4, 0.5, 0.6, 0.7, 0.8, 0.9, 1.0. Note that instead of letting $\beta \to \infty$, the simulation ends with the target distribution at $\beta$ = 1.0. Each temperature is sampled over 20,000 steps.

2.  Selection. The fitnesses of the walkers are determined by their log-posterior values (equation 15). 10 walkers from the best 300 parameter sets sampled in the previous step are chosen and subjected to numerical optimization to find the nearby local maximums, which are then used to seed a sampling run in the next step. In the spirit of elitist selection, the best walker is always included for the next generation.

3.  Recombination and mutation. The initial walkers for the sampling run are generated by adding a Gaussian noise $\mathcal{N}(0, \sigma I)$ to the 10 optimized walkers (here, $I$ is the identity matrix and $\sigma$ = 0.001), and the number of initial walkers centered around each optimized walker $\theta_j$ is given by

$$224 \frac{p(\theta_j|\mathcal{D})^\beta}{\sum_k p(\theta_k|\mathcal{D})^\beta}. \tag{17}$$

That is, the proportion of the initial walkers generated from each optimized walker is weighted by its posterior value with a temperature factor of $\beta$ = 0.6; the temperature factor is chosen to allocate more walkers to optimized walkers with lower posterior values. The sampling run consists of 50,000 steps. Note that the purpose of the Gaussian noise is to ensure that the proposal distribution is valid for any pair of walkers for the ensemble MCMC method (see below), rather than to control the mutation strength, as is done in evolution strategy (Beyer & Schwefel, 2002).

4.  Termination. The best walkers from the sampling run are compared to the optimized walkers. If the best walkers do not

escape to new local maximums with higher posterior values, then the procedure is terminated after an additional 50,000 sampling steps. If, however, new local maximums are discovered during sampling, the algorithm loops back to the selection step. This process is repeated until no better local maximum is discovered at the end of sampling. Unless otherwise specified, only the last 30,000 sampling steps (downsampled every 100 steps) are used for post-analysis.

In general, the number of walkers in ensemble MCMC needs to be larger than the number of free parameters; here, the number 224 is chosen to optimize parallel performance on a local computer cluster (8 nodes × 28 CPU cores/node).

We found that this procedure outperformed either ensemble MCMC or numerical optimization by itself (compare Appendix Fig S2A and B). For the full model, we ran this procedure three times to assess the reproducibility of the fit (see Appendix for further discussion).

### Markov chain Monte Carlo

One major challenge in efficient MCMC sampling in systems biology is that the target distributions are often poorly scaled. In the context of ODE kinetic modeling, this means that different reaction rates and their associated uncertainties can be separated by several orders of magnitude. This is almost certainly true for the KaiA—KaiC subsystem because, among other things, the experimentally measured rates of KaiA binding and dissociation are much faster than the ATP hydrolysis rate of KaiC. Without *a priori* knowledge of the natural time scales, conventional MCMC schemes are inefficient in such sampling problems, because only very small displacements are accepted at appreciable rates. In this work, we employ an ensemble MCMC method developed by Goodman and Weare (2010). The advantage of the Goodman–Weare algorithm is that it is affine invariant, which means that it performs equally well for isotropic and poorly scaled measures, providing that the two can be related by a linear transformation of the coordinate system. This appears to be the case for the present problem since the Goodman–Weare algorithm vastly outperforms a standard Metropolis–Hastings scheme with a (preconditioned) Gaussian proposal distribution (Appendix Fig S2B).

In brief, the Goodman–Weare algorithm evolves an ensemble of walkers, rather than a single one. At each step, individual walker positions are updated sequentially. For a given walker $\theta_k$ at step $\tau$, a walker $\theta_j$ is drawn randomly from the rest of the ensemble and a new position, $\eta$, on the line connecting $\theta_k$ and $\theta_j$ is proposed by a "stretch move"

$$\eta = \theta_j + z\left[\theta_k(\tau) - \theta_j\right] \tag{18}$$

where $z$ is a random number drawn from the distribution

$$Z \sim g(z; \alpha) = \begin{cases} 1/\sqrt{z}, & z \in [1/\alpha, \alpha] \\ 0, & \text{otherwise} \end{cases}. \tag{19}$$

The "stretch factor" $\alpha$ is a tunable parameter that controls the step size. In an *N*-dimensional parameter space, the new walker $\eta$ is accepted with the probability

$$q = \min\left(1, z^{N-1} \frac{p(\eta|\mathcal{D})}{p(\theta_k(\tau)|\mathcal{D})}\right) \tag{20}$$

which guarantees that the scheme obeys detailed balance and thus converges to the target distribution $p(\theta|\mathcal{D})$ as $\tau \to \infty$. Note that no derivative of the posterior distribution is required to draw from the proposal distribution. In this work, we use $\alpha = 1.1$, which gives an average acceptance rate of 47% in steps 3 and 4 of the fitting procedure.

### Numerical optimization

The numerical optimization method used in this work is a modified version of Powell's method (Powell, 1964; Press *et al*, 2007). Briefly, given an initial guess and direction set, which is usually the Cartesian coordinate set, Powell's method performs a line search to minimize the objective function, here $-\ln p(\theta|\mathcal{D})$, sequentially along each vector in the direction set. The direction set is then updated by replacing the direction of largest decrease in the objective function in the current iteration with the displacement vector from the estimated minimum at the beginning to that at the end of the line minimizations, provided that certain technical conditions are met to avoid the buildup of linear dependence in the direction set. This process is repeated until a convergence threshold is met. Note that unlike the original method, the modified Powell's method does not guarantee that the vectors in the direction set are mutually conjugate.

Similar to the Goodman–Weare algorithm, Powell's method is derivative-free. For the current system, Powell's method converges faster than the Nelder–Mead method (Nelder & Mead, 1965), another commonly used derivative-free method, although the Nelder–Mead method appears less prone to becoming trapped in local metastable states (Appendix Fig S2C).

### Software implementation

The fitting procedure is implemented in an in-house Python script that interfaces with several existing Python modules: Numerical integration of the model ODEs is done using the Odespy package (Langtangen & Wang, 2015) with the BDF method; the Goodman–Weare algorithm is implemented in emcee (version 2.2.1) (Foreman-Mackey *et al*, 2013); Powell's method and the Nelder–Mead method are implemented in SciPy (version 1.2.1) (Virtanen *et al*, 2020). The derivative evaluation step in ODE integration is accelerated using numba (Lam *et al*, 2015), and the script is parallelized using mpi4py 2.0.0 (Dalcin *et al*, 2011; Dalcín *et al*, 2005, 2008).

The most computationally expensive step in the fitting procedure is the MCMC sampling, because each move requires multiple ODE evaluations to compute the posterior function. With 224 walkers and eight nodes (each with 28 Intel E5-2680v4 2.4 GHz cores), the speed of MCMC sampling is 46,000 steps per hour.

### Model comparison

In the preceding sections, all definitions of probability distributions implicitly assume that there is a model $\mathcal{M}$ with a well-defined functional form, whose parameters $\theta$ need to be determined. For the sake of model comparison, we make this assumption explicit and rewrite (11) as

$$p(\theta|\mathcal{D}, \mathcal{M}) = \frac{\mathcal{L}(\mathcal{D}|\theta, \mathcal{M})p(\theta|\mathcal{M})}{p(\mathcal{D}|\mathcal{M})}. \tag{21}$$

To compare two models $\mathcal{M}_i$ and $\mathcal{M}_j$, we need to compare the posterior probabilities for each model, usually in the form of their ratios

$$\frac{p(\mathcal{M}_i|\mathcal{D})}{p(\mathcal{M}_j|\mathcal{D})} = \frac{p(\mathcal{D}|\mathcal{M}_i)p(\mathcal{M}_i)}{p(\mathcal{D}|\mathcal{M}_j)p(\mathcal{M}_j)}. \tag{22}$$

Assuming that we have no prior preference for any model, the ratio becomes the Bayes factor

$$B_{ij} = \frac{p(\mathcal{D}|\mathcal{M}_i)}{p(\mathcal{D}|\mathcal{M}_j)}, \tag{23}$$

which we adopt as the metric for model comparison.

The primary difficulty in computing the Bayes factor is thus estimating the evidence, or the marginal likelihood function, for each $\mathcal{M}_i$. There are several methods for computing the evidence (Vyshemirsky & Girolami, 2008). Here, we derive a formula compatible with the ensemble MCMC scheme that is directly analogous to free energy perturbation (Zwanzig, 1954). For the sake of simplicity, we drop the model index $i$ from this point on. First, note that for a given model $\mathcal{M}$,

$$p(\mathcal{D}|\mathcal{M}) = \int \mathcal{L}(\mathcal{D}|\theta,\mathcal{M})p(\theta|\mathcal{M}) \, d\theta = \langle \mathcal{L}(\mathcal{D}|\theta,\mathcal{M}) \rangle_{p(\theta|\mathcal{M})} \tag{24}$$

where the first equality follows from the law of total probability and the second equality assumes that the prior $p(\theta|\mathcal{M})$ is normalized (as a probability density function of $\theta$). Equation (24) suggests that the marginal likelihood function can be computed by estimating the average of the likelihood function $\mathcal{L}(\mathcal{D}|\theta,\mathcal{M})$ against the prior. Using MCMC to estimate this integral is inefficient since there is very little overlap between the likelihood function and the prior for the models of interest. Instead, we define

$$q_\lambda(\theta) = \mathcal{L}(\mathcal{D}|\theta,\mathcal{M})^\lambda p(\theta|\mathcal{M}) \quad \text{and} \quad Z_\lambda = \int q_\lambda(\theta) \, d\theta$$

for $0 \leq \lambda \leq 1$ and then note that (24) can be recast as

$$p(\mathcal{D}|\mathcal{M}) = \frac{Z_1}{Z_0} = \left( \frac{Z_{\lambda_0}}{Z_{\lambda_1}} \frac{Z_{\lambda_1}}{Z_{\lambda_2}} \cdots \frac{Z_{\lambda_{N-1}}}{Z_{\lambda_N}} \right)^{-1}. \tag{25}$$

For $0 = \lambda_0 < \lambda_1 < \cdots < \lambda_N = 1$, and the $\lambda$s are chosen to allow for sufficient overlap between successive $q_\lambda(\theta)$s. Each fraction in (25) is given by

$$\frac{Z_{\lambda_{n-1}}}{Z_{\lambda_n}} = \frac{\int \mathcal{L}(\mathcal{D}|\theta,\mathcal{M})^{\lambda_{n-1}-\lambda_n} q_{\lambda_n}(\theta) \, d\theta}{\int q_{\lambda_n}(\theta) \, d\theta} = \left\langle \mathcal{L}(\mathcal{D}|\theta,\mathcal{M})^{\lambda_{n-1}-\lambda_n} \right\rangle_{q_{\lambda_n}}. \tag{26}$$

Therefore,

$$p(\mathcal{D}|\mathcal{M}) = \prod_{n=1}^{N} \left\langle \mathcal{L}(\mathcal{D}|\theta,\mathcal{M})^{\lambda_{n-1}-\lambda_n} \right\rangle_{q_{\lambda_n}}^{-1} = \prod_{n=1}^{N} \left\langle e^{\ln q_{\lambda_{n-1}}(\theta) - \ln q_{\lambda_n}(\theta)} \right\rangle_{q_{\lambda_n}}^{-1} \tag{27}$$

where the averages $\langle \cdot \rangle_{q_{\lambda_n}}$ can be approximated with MCMC. Equation (27) is a version of the free energy perturbation formula.

Note that (27) requires that the likelihood function $\mathcal{L}(\mathcal{D}|\theta,\mathcal{M})$ be properly normalized (as a probability density function of $\mathcal{D}$), but does not require the prior $p(\theta|\mathcal{M})$ to be normalized, as any missing normalization constant cancels out in each term of the product.

For each simplified model in Table 1 and Appendix Table S1, the ensemble of walkers from the last time step of the model fitting procedure is used to initialize an MCMC sampling run with $\lambda = 1.00$. The lambda value is reduced by 0.01 at each subsequent stage until $\lambda$ reaches 0.01. Each stage is sampled for 2,000 time steps using the Goodman–Weare ensemble sampler (Goodman & Weare, 2010). Only the last 1,000 time steps from each stage is used to compute the ensemble average in (27). The Bayes factors are then computed as the ratios of the evidence for the full model to each simplified model.

### Refitting

The steady-state KaiC phosphorylation measurements (Figs 3B and EV4B) are fit to the full model using Powell's method, starting from the best fit based on the training data set. The priors on the kinetic parameters (Table 2) are centered on the best fit values, so that the refit model can be interpreted as the "minimal" perturbation to the best fit that enables agreement with the steady-state measurement.

### Curve fitting

The experimentally determined stimulus–response relations for KaiC S431A (Fig 3C and F) are fit to the simple inhibitor ultrasensitivity scheme described in Box 5 of Ferrell and Ha (2014b). Using their notation, the amount of phosphorylated protein substrate (%XP) as a function of kinase concentration ([$K$]) is given by

$$\%XP([K])$$
$$= P_{\max} \frac{K_1[I] + K_1 K_2 - K_1[K] + 2K_2[K] - K_1\sqrt{[I]^2 + 2(K_2 - [K])[I] + (K_2 + [K])^2}}{2K_1[I] - 2(K_1 - K_2)(K_1 + [K])}$$
$$+ b. \tag{28}$$

Here, $P_{\max}$, [$I$], $K_1$, $K_2$, and $b$ are free model parameters. Unlike the Hill function, EC50 is not an explicit parameter of (28) and thus needs to be determined numerically. Note that equation (28) can be reduced to a right-shifted hyperbolic function as $K_2 \to 0$:

$$\%XP([K]) = P_{\max} \frac{[I] - [K] - |[I] - [K]|}{2[I] - 2(K_1 + [K])} + b, \tag{29}$$

which is equivalent to a threshold–hyperbolic stimulus–response function,

$$\%XP([K]) = \frac{P_{\max}([K] - [I])}{K_1 + ([K] - [I])} H([K] - [I]) + b, \tag{30}$$

where $H$ is the unit step function.

Stimulus–response relations are fit using the NonlinearModelFit function in Mathematica 12.0. We stress here that the curve fits are purely phenomenological and are thus not intended to be interpreted in terms of the biochemical assumptions underlying the model.

### Numerical method for the Phong model

All simulations of the modified Phong model (Figs 4E and EV3E and F) were carried out in Mathematica 12.0 using the NDSolve function. To determine the period of the model for a given parameter set, the model was simulated for 200 h and the first 100 h was discarded to eliminate transient responses. The NDSolve function returns ODE solutions as InterpolatingFunction objects, which were converted to time series sampled at every 0.1 h for subsequent analyses. The troughs in the time series of the total KaiC phosphorylation level (i.e., $[C^P] = [C^T] + [C^S] + [C^D] + [^BC^S] + [^BC^D]$) were determined using the FindPeaks function, and the average of the trough-to-trough time was taken to be the period of the model. The model was considered non-oscillatory if one of the following conditions was met:

1  The standard deviation of the $[C^P]$ time series is $< 0.1\ \mu M$ (weak oscillation).
2  The standard deviation of the trough amplitude is $> 0.001\ \mu M$ (damped oscillation).
3  The number of troughs is $< 3$ (abnormally long period).

## Experimental methods

### Protein expression and purification

KaiA, KaiB, KaiB-FLAG, and KaiC were expressed and purified as previously described (Phong et al, 2013) with two modifications to the protocol: Anion exchange chromatography was performed using HiTrap Q columns (GE Healthcare), and KaiC was purified using Ni-NTA affinity chromatography followed by size-exclusion chromatography, omitting the anion exchange step. The expression, purification, and 6-iodoacetofluorescein (6-IAF) labeling of KaiB K25C mutant as a fluorescence reporter in the plate reader assay is described in Leypunskiy et al (2017). All mutants of KaiC were constructed using QuikChange II XL Site-Directed Mutagenesis Kit (Agilent). For the KaiC AA and KaiC-EE mutants, the His-tags were not cleaved during purification; this ensures that these mutant proteins have shifted mobility in SDS–PAGE, allowing their bands to separate from those of KaiC S431A.

The U-[$^{15}$N] labeled N-terminal (residues 1–135) and C-terminal (residues 181–284) domains of KaiA were expressed in BL21(DE3) Escherichia coli (Novagen) in minimal (M9) media supplemented with $^{15}$N-enriched NH$_4$Cl. For the expression of the C-terminal domain, M9 media enriched with $^{15}$N-NH$_4$Cl was prepared using 98% deuterated water (D$_2$O). The proteins were purified by Ni-NTA affinity chromatography followed by size-exclusion chromatography using a Superdex 75 1660 prep grade column, as described previously (Chang et al, 2011, 2012; Tseng et al, 2014). N-terminal KaiA eluted as a ~15 kDa monomer (Vakonakis et al, 2001), while C-terminal KaiA eluted as a ~23 kDa homodimer (Vakonakis & LiWang, 2004).

GFP was expressed as a N-terminal 6xHis-tag fusion from the pET28a plasmid in the BL21 (DE3) strain of E. coli. Harvested cells were sonicated for lysis, and clarified lysate was loaded onto a HisTrap FF column (GE Healthcare). The His tag was cleaved by overnight incubation at 4°C with SUMO protease (Invitrogen), after which the sample was loaded again onto a HisTrap FF column to recover the cleaved products. The cleaved proteins were further purified on a 5 ml HiTrap Q HP column (GE Healthcare) and then a Superdex 200 10/300 GL size-exclusion column. The eluted

fractions were concentrated in a sample buffer (20 mM HEPES [pH 7.4], 150 mM KCl, 2.5 mM MgCl$_2$, 2 mM DTT), aliquoted, and snap-frozen in liquid nitrogen for storage at −80°C.

### In vitro clock reactions

All in vitro clock reactions were done in the standard reaction buffer (20 mM Tris–HCl [pH 8], 150 mM NaCl, 5 mM MgCl$_2$, 0.5 mM EDTA, 10% glycerol, 50 μg·ml$^{-1}$ Kanamycin). KaiC concentration was 3.5 μM in all experiments unless otherwise specified; KaiB concentration was 3.5 μM for all oscillatory reactions, and 6-IAF-labeled KaiB K25C concentration was 0.2 μM for plate reader assays. KaiA concentration and %ATP were determined by each individual experiment, while the total nucleotide concentration (i.e., [ATP] + [ADP]) was held constant at 5 mM. Phosphorylation kinetics was resolved using SDS–PAGE on 10% acrylamide gels (37.5:1 acrylamide:bis-acrylamide) run for 4.5 h at 30 mA constant current at 12°C; the gels were stained in SimplyBlue SafeStain (Invitrogen) and then imaged using Bio-Rad ChemiDoc Imager. The oscillatory reactions (Figs 2D and EV3C) were also monitored using the plate reader assay described in Leypunskiy et al (2017).

### NMR spectroscopy

A Bruker 600 MHz AVANCE III spectrometer equipped with a TCI cryoprobe was used for all of the NMR experiments of the N- and C-terminal domains of KaiA (Appendix Fig S1). Chemical shifts were referenced to internal 2,2-dimethyl-2-silapentane-5-sulfonate (10 μM). Data were processed using NMRPipe and visualized using NMRDraw (Delaglio et al, 1995). NMR samples were prepared with 100 μM monomer concentration of protein in 20 mM Tris–HCl [pH 8], 150 mM NaCl, 5 mM MgCl$_2$, and 5% D$_2$O buffer. All experiments were performed at 30°C. Samples were incubated with 1 mM ATP or ADP, when needed, for 30 min before spectral measurement.

### Immunoprecipitation

Immunoprecipitation of KaiB-FLAG and associated protein complexes in a clock reaction (Fig 4A) was done as previously described (Phong et al, 2013) using monoclonal anti-FLAG M2 antibody (Sigma-Aldrich, product number F1804), and elution with 3xFLAG peptide. The supernatant was analyzed by SDS–PAGE on 4–20% Criterion TGX Stain-Free Precast Gels (Bio-Rad) and stained with SYPRO Ruby (Bio-Rad). 1.5 μM GFP was added to the reaction mixture at the beginning of the time course and served as an internal standard to correct for changes in protein concentration due to handling. The relative supernatant KaiA concentration was determined as a ratio of KaiA band intensity in each lane to the GFP band intensity and is normalized as percentage of the largest ratio in the time course (Fig EV3D).

## Data and software availability

The code used to perform and analyze the MCMC simulations, as well as data from the inference run (shown in blue in Fig EV1B), is available on GitHub at https://github.com/luhong88/KaiAC_MCMC. The parameters for the inference run are available in Appendix Tables S2 and S3.

**Expanded View** for this article is available online.

## Acknowledgements

We thank Connie Phong and Haneul Yoo for their protein samples, Jonathan Weare for helpful discussions, and Steven Redford for critical readings of the manuscript. This work was supported by National Science Foundation award MCB-1953402, National Institutes of Health awards GM107369, GM107521, GM135382, and EY025957, Department of Energy Office of Advanced Scientific Computing Research contract DE-AC02-06CH11347 and award DE-SC0014205, and a Howard Hughes Medical Institute-Simons Foundation Faculty Scholarship (to MJR). AL was also supported by the Center for Cellular and Biomolecular Machines at University of California, Merced (NSF Grant HRD-1547848). Computations were performed on resources provided by the University of Chicago Research Computing Center, and the Extreme Science and Engineering Discovery Environment (Towns *et al*, 2014) (NSF Grant ACI-1548562) Bridges (PSC) computing nodes through allocation TG-MCB180007.

## Author contributions

LH, ELe, MJR, and ARD conceptualized the project; DOL, ELe, LH, and AC performed experiments; DOL, LH, and ELi developed software; LH and DOL performed computational analyses; ARD, MJR, AL, and CM provided supervision; LH, MJR, and ARD wrote and edited the paper.

## Conflict of interest

The authors declare that they have no conflict of interest.

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
