## [Review Process File · Molecular Systems Biology]

Bayesian modeling reveals metabolite-dependent ultrasensitivity in the cyanobacterial circadian clock

Lu Hong, Danylo Lavrentovich, Archana Chavan, Eugene Leypunskiy, Eileen Li, Charles Matthews, Andy LiWang, Michael Rust, and Aaron Dinner

DOI: [10.15252/msb.20199355](https://doi.org/10.15252/msb.20199355)

Corresponding author(s): Aaron Dinner (dinner@uchicago.edu) Contributing Author Michael Rust with email address mrust@uchicago.edu has been identified as a secondary point of correspondence if this manuscript is accepted for publication.

Review Timeline:

Submission Date:	12th Nov 19
Editorial Decision:	19th Dec 19
Revision Received:	28th Feb 20
Editorial Decision:	9th Apr 20
Revision Received:	21st Apr 20
Accepted:	24th Apr 20

Editor: Jingyi Hou

Transaction Report:

19th Dec 2019

Manuscript Number: MSB-19-9355

Title: Bayesian Modeling Reveals Ultrasensitivity Underlying Metabolic Compensation in the Cyanobacterial Circadian Clock

Author: Lu Hong

Danylo Lavrentovich

Archana Chavan

Eugene Leypunskiy

Eileen Li

Charles Matthews

Andy LiWang

Michael Rust

Aaron Dinner

Dear Prof Dinner,

Thank you for submitting your work to Molecular Systems Biology. We have now heard back from the three referees who agreed to evaluate your study. As you will see below, the reviewers acknowledge that the presented findings seem potentially interesting. They raise however a series of concerns, which we would ask you to address in a major revision.

I think that the reviewers' recommendations are rather clear and there is therefore no need to repeat the comments listed below. In light of the concerns of Reviewer #3, we would ask you to edit the manuscript to make sure that the main findings are sufficiently clear and easily accessible to the general audience of Molecular Systems Biology.

All other issues raised by the reviewers need to be satisfactorily addressed. As you may already know, our editorial policy allows in principle a single round of major revision and it is therefore essential to provide responses to the reviewers' comments that are as complete as possible. Please feel free to contact me in case you would like to discuss in further detail any of the issues raised by the reviewers.

On a more editorial level, we would ask you to address the following issues:

- Please provide a .docx formatted version of the manuscript text (including legends for main figures, EV figures and tables). Please make sure that the changes are highlighted to be clearly visible.
- Please provide individual production quality figure files as .eps, .tif, .jpg (one file per figure).
- Please provide a .docx formatted letter INCLUDING the reviewers' reports and your detailed point-by-point responses to their comments. As part of the EMBO Press transparent editorial process, the point-by-point response is part of the Review Process File (RPF), which will be published alongside your paper.
- Please note that all corresponding authors are required to supply an ORCID ID for their name upon

submission of a revised manuscript.

-We replaced Supplementary Information with Expanded View (EV) Figures and Tables that are collapsible/expandable online (see examples in <http://msb.embopress.org/content/11/6/812>). A maximum of 5 EV Figures can be typeset. EV Figures should be cited as 'Figure EV1, Figure EV2' etc... in the text and their respective legends should be included in the main text after the legends of regular figures.

- For the figures that you do NOT wish to display as Expanded View figures, they should be bundled together with their legends in a single PDF file called *Appendix*, which should start with a short Table of Content. Appendix figures should be referred to in the main text as: "Appendix Figure S1, Appendix Figure S2" etc. See detailed instructions regarding expanded view here: <https://www.embopress.org/page/journal/17444292/authorguide#expandedview>.

- Before submitting your revision, primary datasets (and computer code, where appropriate) produced in this study need to be deposited in an appropriate public database (see <https://www.embopress.org/page/journal/17444292/authorguide#dataavailability>). - Dataset #1
- Dataset #2>

The accession numbers and database should be listed in a formal "Data Availability " section (placed after Materials & Method) that follows the model below (see also <https://www.embopress.org/page/journal/17444292/authorguide#dataavailability>). Please note that the Data Availability Section is restricted to new primary data that are part of this study.

Data availability

- We would encourage you to include the source data for figure panels that show essential quantitative information. Additional information on source data and instruction on how to label the files are available at (<https://www.embopress.org/page/journal/17444292/authorguide#sourcedata>).

- All Materials and Methods need to be described in the main text. We would encourage you to use 'Structured Methods', our new Materials and Methods format. According to this format, the Material and Methods section should include a Reagents and Tools Table (listing key reagents, experimental models, software and relevant equipment and including their sources and relevant

identifiers) followed by a Methods and Protocols section in which we encourage the authors to describe their methods using a step-by-step protocol format with bullet points, to facilitate the adoption of the methodologies across labs. More information on how to adhere to this format as well as downloadable templates (.doc or .xls) for the Reagents and Tools Table can be found in our author guidelines: <

<https://www.embopress.org/page/journal/17444292/authorguide#researcharticleguide>>. An example of a Method paper with Structured Methods can be found here: .

- Please provide a "standfirst text" summarizing the study in one or two sentences (approximately 250 characters, including space), three to four "bullet points" highlighting the main findings and a "synopsis image" (550px width and max 400px height, jpeg format) to highlight the paper on our homepage.

- When you resubmit your manuscript, please download our CHECKLIST (http://embopress.org/sites/default/files/Resources/EP_Author_Checklist.xls) and include the completed form in your submission. *Please note* that the Author Checklist will be published alongside the paper as part of the transparent process <http://msb.embopress.org/authorguide#transparentprocess>.

If you feel you can satisfactorily deal with these points and those listed by the referees, you may wish to submit a revised version of your manuscript. Please attach a covering letter giving details of the way in which you have handled each of the points raised by the referees. A revised manuscript will be once again subject to review and you probably understand that we can give you no guarantee at this stage that the eventual outcome will be favorable.

Yours sincerely,

Jingyi Hou
Editor
Molecular Systems Biology

If you do choose to resubmit, please click on the link below to submit the revision online *within 90 days*.

Link Not Available

IMPORTANT: When you send your revision, we will require the following items:

1. the manuscript text in LaTeX, RTF or MS Word format
2. a letter with a detailed description of the changes made in response to the referees. Please specify clearly the exact places in the text (pages and paragraphs) where each change has been made in response to each specific comment given
3. three to four 'bullet points' highlighting the main findings of your study
4. a short 'blurb' text summarizing in two sentences the study (max. 250 characters)
5. a 'thumbnail image' (width=211 x height=157 pixels, Illustrator, PowerPoint, OmniGraffle or jpeg format), which can be used as 'visual title' for the synopsis section of your paper.

6. Please include an author contributions statement after the Acknowledgements section (see <https://www.embopress.org/page/journal/17444292/authorguide>)
7. Please complete the CHECKLIST available at (<http://bit.ly/EMBOPressAuthorChecklist>). Please note that the Author Checklist will be published alongside the paper as part of the transparent process (<https://www.embopress.org/page/journal/17444292/authorguide#transparentprocess>).
8. Please note that corresponding authors are required to supply an ORCID ID for their name upon submission of a revised manuscript (EMBO Press signed a joint statement to encourage ORCID adoption). (<https://www.embopress.org/page/journal/17444292/authorguide#editorialprocess>)

Currently, our records indicate that there is no ORCID associated with your account.

Please click the link below to provide an ORCID:

Link Not Available

The system will prompt you to fill in your funding and payment information. This will allow Wiley to send you a quote for the article processing charge (APC) in case of acceptance. This quote takes into account any reduction or fee waivers that you may be eligible for. Authors do not need to pay any fees before their manuscript is accepted and transferred to the publisher.

*** PLEASE NOTE *** As part of the EMBO Press transparent editorial process initiative (see our Editorial at <http://dx.doi.org/10.1038/msb.2010.72>), Molecular Systems Biology publishes online a Review Process File with each accepted manuscripts. This file will be published in conjunction with your paper and will include the anonymous referee reports, your point-by-point response and all pertinent correspondence relating to the manuscript. If you do NOT want this File to be published, please inform the editorial office at msb@embo.org within 14 days upon receipt of the present letter.

Reviewer #1:

In this detailed work, the authors carefully examine the cyanobacterial circadian clock mechanism using a combination of experiment and modelling, with optimization of the model carried out using bayesian methods. It is a strength of the work that the authors are able to break down both the experimental system (using a well established invitro assay of the clock) and the model to focus on key parameters and interactions. The authors find ultrasensitivity in KaiC phosphorylation as a function of KaiA, which they find evidence for experimentally. The authors go onto propose that this ultrasensitivity in KaiC phosphorylation plays a role in stabilizing the oscillator at low ATP conditions by blocking phosphorylation occurring during the dephosphorylation stage. This represents an interesting new step forward in our understanding of the mechanism of the cyanobacterial clock, which will be of interest to circadian and systems biologists.

The one concern I have is that I could not easily ascertain how significant the ultrasensitivity revealed in KaiC phosphorylation as a function of KaiA is for the mechanism of clock oscillations. As the authors describe, multiple models have been made of the cyanobacterial clock, with multiple

aspects of the network capable of generating the cooperativity/ultrasensitivity required for oscillations. These aspects include the hexameric structure of the KaiC protein, as well as the transcriptional feedback loop formed due to KaiC binding its own promoter. The authors describe these aspects of the network as other aspects that can promote cooperativity, and suggest that 'the presence of nonlinearities and delayed feedback at multiple steps in a molecular oscillator allows the system to achieve greater robustness'. Is there anyway the authors can further estimate the strength, or importance for the generation of oscillations, of the ultrasensitivity in KaiC phosphorylation in KaiA that they measure? It was unclear to me whether this ultrasensitivity alone was enough to generate robust oscillations (sorry if I missed this). It could also be possible to compare the strength of this ultrasensitivity to the previously proposed 'ultrasensitivity in KaiB-dependent KaiA sequestration that arises from opposing S and T phosphorylations 524 within hexamers (Lin et al., 2014)', or the transcriptional feedback loop due to KaiC regulating its own promoter. It is great that the authors find that the addition of ultrasensitivity to a previously generated model can increase the range of conditions where oscillations are possible, which is a step in this direction.

Reviewer #2:

-Summary

Hong et al. analyzed the KaiC phosphorylation kinetics based on the Bayesian modeling and proposed a model to explain the robust oscillation against the ATP/ADP alteration, which is called the metabolic compensation. Ever since the reconstitution of the circadian rhythm of KaiC phosphorylation in vitro, the mathematical modeling approach has been applied to understand the robust oscillation of KaiC phosphorylation dynamics. In this study, the authors performed the data-driven approach instead of the forward modeling approach. Although KaiC protein is a homo-hexamer and each monomer has CI and CII domains, they modeled only a CII domain to enable the data-driven approach. After constructing the biochemical model of the CII domain, they sampled biochemical parameters to fit the biochemical experiments. From the sampled parameters, they found the unphosphorylated CII-ADP complex has a high affinity to KaiA. The authors also found the competition of the binding between different-KaiC states and KaiA generates the ultrasensitivity of the KaiC phosphorylation with respect to the KaiA concentration. The mechanism underlying the ultrasensitivity can be categorized as a substrate competition model. Based on these findings, they introduced the ultrasensitivity to the simplified oscillation model previously developed by the authors. By the modified mathematical model, the authors confirmed that the introduced ultrasensitivity increases the robustness of the oscillation by changing the ATP/ADP ratio.

-General remarks

The description of the Bayesian modeling is detailed and the model is well fitted to the experimental data. The computation of the Bayes factor for each model is useful to compare different models in a quantitative way. Thus, the current study has an impact on quantitative modeling in the field of computational biochemistry. The finding of ultrasensitivity in the action of KaiA to KaiC and the mechanistic insight underlying it are important to the circadian field as well as general biochemistry. However, I have some issues to be addressed as follows:

-Major points

1. The authors write the section title "Substrate competition underlies metabolic compensation" in line 417 and the title "Ultrasensitivity Underlying Metabolic Compensation". This seems to be the overstatement because the authors do not perform any experiments to validate the mathematical model in Fig. 4. To state "underlies", the author should perform the verification experiment. In Fig. 3F, the authors showed KaiC-AA mutant can mimic the substrate competition by the unphosphorylated KaiC. Is the metabolic compensation affected or invariant by titrating KaiC-AA or KaiC-EE mutants in the in vitro oscillation experiment? Because the authors interpret the ultrasensitivity as the synchronization step, it should be also interesting to observe the amplitude change of reconstituted oscillation in the presence of KaiC-AA.

2. In Eq. (38), the ultrasensitivity is introduced by hand. Since the authors found the substrate competition mechanism generates the ultrasensitivity, it seems to be possible to derive the kinetic equation like Eq. (28). At least it is requested to discuss whether the strength of observed KaiC(AA)-EC50 relationship (Fig. 3G) is reasonable to the level of ultrasensitivity estimated by the authors' model.

-Minor points

1. In Fig. 1D, dissociation constants are denoted as K_d^{DP} or K_d^{TP} . This notation cannot discriminate the phosphorylation states. Since the authors use the notation $k_a^{\{U,DP\}}$ for the rate constant, it seems to be better to use $K_d^{\{X,DP\}}$ instead of K_d^{DP} . Then, it is helpful to show labels such as X = U, T, S, D with corresponding colors in the inset of the figure.

2. In Fig. 3E, the labels "threshold" and "switch" are ambiguous. They should be replaced with EC10 and EC90 - EC10, respectively.

3. In Fig. 4G, what is represented by orange-rectangles in the background?

4. In Fig. S1B, S4DF, and S10B, labels, and units of the x-axis are missing.

5. In Fig. S1B, distributions of $k_{TP}^{\{A,X\}}$ are presented in "Nuc. exch.". To my understand, these parameters depend on %ATP as $k_r^{\{DP,X\}} * [ATP]/([ATP] + [ADP])$ from Eq. (4). If so, %ATP-independent parameters $k_r^{\{DP,X\}}$ should be plotted instead of $k_{TP}^{\{A,X\}}$.

6. In the legend of Fig. S4, it is helpful to cite references of the original experimental data.

7. In Fig. S6, the notation $(dC/C)/(dk/k)$ in x and y labels is not correct. It should be replaced with $d\log(C)/d\log(k)$ or $(dC/dk)/(C/k)$.

8. Since Fig. S7A is a different representation of Fig. 2D, Fig. S7A can be removed. Then, it seems to be readable to move Fig. S7B to the next of Fig. 2D with the scatter plot similar to Fig. 2D.

9. Typo in line 824. Aligent should be Agilent.

10. In line 847, the authors wrote "(i.e., $[ATP] + [ADP]$) was held constant at 5 μ M.". I guess the unit of concentration is not μ M but mM.

11. In Table 4, it is better to show the absolute concentrations of ATP and ADP instead of %ATP.

12. I could not find the numerical method to solve the modified Phong model and the analysis

method to compute the period shown in Fig. 4E. It is appropriate to describe the method in or after the section "Phenomenological modifications to the Phong model" (line 1069).

Reviewer #3:

Here, the authors use a Bayesian parameter estimation approach to uncover metabolic mechanisms underlying the cyanobacterial circadian clock. Specifically, they investigate the daytime interactions among the various phosphorylation and nucleotide-bound states of KaiC and KaiA. First, they use the results from parameter estimation to illustrate the transient kinetic states of the KaiC protein. In particular, their model replicates the KaiC preference of phosphorylation at the T site before phosphorylation at the S site. Using their model, they propose that the preference is more than just a difference in the relative unphosphorylated to T-phosphorylated and unphosphorylated to S-phosphorylated reaction rates. They also show that the ADP-bound states of KaiC are unstable relative to the ATP-bound states. Next, they show that the KaiA concentration, to an extent, mitigates the clock's sensitivity and sensitivity of the period to the percent of ATP present. Their analysis reveals, however, that KaiA concentration is not sufficient to fully abolish the sensitivity of the clock to percent ATP.

The remaining results discuss the ultrasensitivity of KaiC phosphorylation on KaiA levels. The authors appeal to previous work on ultrasensitivity to justify their claim that KaiC phosphorylation is ultrasensitive to KaiA binding affinity to the nucleotide state of unphosphorylated KaiC. See below for two main criticisms regarding this conclusion and how the authors can address them. The authors argue that the mechanism is substrate competition, where the competition is driven by differential binding to multiple substrates. They corroborate this hypothesis through two experiments. First, they measure the stimulus-response relation of KaiC S431A mutant in the presence of both KaiC S431A/T432A and KaiCS431E/T432E. Each mutant acts as a competitor to KaiA-KaiC interaction. Second, they show that the model exhibits weak nonlinearity when KaiA has near equivalent binding affinity for the two nucleotide-bound states. To show this, they use a Bayes factor calculation revealing that near equivalent binding affinity significantly reduces the quality of the fit. In contrast, a model with differential binding affinities for the nucleotide-bound state with equivalent affinity across phosphorylation states has little effect on the fit quality under the Bayes factor.

Finally, the authors show that ultrasensitivity leads to metabolic compensation of the full oscillator. Previous models of the KaiABC oscillator require a high binding affinity between KaiB and KaiA, leading to sequestration of KaiA, to exhibit oscillations. However, the authors show that coupling the protein sequestration with the ultrasensitivity leads to more robust oscillations. They provide experimental evidence that protein sequestration is not solely responsible for oscillations and give a compelling description of how the ultrasensitivity underlies metabolic compensation.

The cyanobacterial is a well-studied model organism in the field of chronobiology with many of the mechanisms underlying the core clock well understood. The authors advance the field using *in silico* and experimental methods to uncover an important aspect to the generation of oscillations in cyanobacteria, namely the ultrasensitivity of KaiC phosphorylation to differential binding affinities of KaiA. The result is significant in that it challenges the current view that sequestration of KaiA by KaiBC provides sufficient nonlinearity in the system to generate oscillations. The study will be of interest to both circadian and computational biologists.

Major Concerns

1) While I like the Bayesian modeling approach and the data specifically collected to test the model, I think the manuscript overemphasized this approach in that I don't think the manuscript will be read because of the Bayesian approach. In my mind, what the manuscript says about the mechanism of cyanobacterial timekeeping is of sufficient interest to readers, and the approach they use is not that different than what is found in some other models. So, while I think it is good to highlight the approach as a strength of the manuscript, it seems overemphasized.

2) The reader has to wade through quite a number of details until the most interesting aspects of the manuscript (e.g., about sequestration) are presented. I think that some reorganization or rewording could make the manuscript more approachable to a general audience. In general, the writing is good, but I do think that, written the right way, the main biological takeaways could be more easily apparent. At the moment the manuscript reads more as would a chronological description of what happened and I think jumping in the text to some more of the key points may be helpful.

3) One clarification concerns the conclusion of the ultrasensitivity dependence of KaiC phosphorylation on KaiA levels. The authors use the phrase "threshold-hyperbolic stimulus-response relation" and cite the work of Gomez-Urbe from 2007. In that work, however, the authors identify four distinct responses of steady-state response. The third response is a "threshold-hyperbolic stimulus-response function", which is a different response than the fourth response, ultrasensitivity.

4) Next, the authors conclude ultrasensitivity using a metric from previous work by Gunawardena. In particular, they calculate EC90-EC10 where EC90 (EC10) is the KaiA concentration required to reach 90% (respectively, 10%) of the steady-state phosphorylation level at saturation. However, in his work from 2005, Gunawardena uses the cooperativity index defined as EC_{90}/EC_{10} . The value EC_{90}/EC_{10} was used in the original work on ultrasensitivity by Goldbeter and is the canonical measure of ultrasensitivity still. The authors should normalize their response curves and recalculate their cooperativity index using EC_{90}/EC_{10} and compare the values to that of the defined bound of ultrasensitivity, namely 81. Or, at least, they should note that their metric is different from that of Gunawardena and justify its use.

5) Mass action should be hyphenated when used as an adjective (as in mass-action kinetics on lines 140, 351, and in the caption of figure 1).

6) It would be nice near the derivation of Eqn. (36) in the Supplemental Information to point out its similarities to the work of Kim and Forger, 2012.

7) The authors italicize *apo* in the Supplemental Information but not in the main text on line 167.

Point-by-point response to reviewer comments for Manuscript Number: MSB-19-9355

We thank the reviewers for their careful reading of our manuscript and their suggestions. All three characterized the discovery of ultrasensitivity as important and said the manuscript will be of interest to both circadian and systems biologists. Reviewers #1 and #2 asked for clarifications regarding the strength of the ultrasensitivity and its impact on the dynamics, and Reviewer #2 asked for further experiments to test the role of ultrasensitivity in metabolic compensation; Reviewers #2 and #3 additionally made a number of suggestions for how best to present our work. We have tried to address all of the comments carefully, as detailed below. Reviewers' comments are reproduced in black, and our responses are in blue.

Reviewer #1:

In this detailed work, the authors carefully examine the cyanobacterial circadian clock mechanism using a combination of experiment and modelling, with optimization of the model carried out using bayesian methods. It is a strength of the work that the authors are able to break down both the experimental system (using a well established in vitro assay of the clock) and the model to focus on key parameters and interactions. The authors find ultrasensitivity in KaiC phosphorylation as a function of KaiA, which they find evidence for experimentally. The authors go onto propose that this ultrasensitivity in KaiC phosphorylation plays a role in stabilizing the oscillator at low ATP conditions by blocking phosphorylation occurring during the dephosphorylation stage. This represents an interesting new step forward in our understanding of the mechanism of the cyanobacterial clock, which will be of interest to circadian and systems biologists.

The one concern I have is that I could not easily ascertain how significant the ultrasensitivity revealed in KaiC phosphorylation as a function of KaiA is for the mechanism of clock oscillations. As the authors describe, multiple models have been made of the cyanobacterial clock, with multiple aspects of the network capable of generating the cooperativity/ultrasensitivity required for oscillations. These aspects include the hexameric structure of the KaiC protein, as well as the transcriptional feedback loop formed due to KaiC binding its own promoter. The authors describe these aspects of the network as other aspects that can promote cooperativity, and suggest that 'the presence of nonlinearities and delayed feedback at multiple steps in a molecular oscillator allows the system to achieve greater robustness'. Is there anyway the authors can further estimate the strength, or importance for the generation of oscillations, of the ultrasensitivity in KaiC phosphorylation in KaiA that they measure? It was unclear to me whether this ultrasensitivity alone was enough to generate robust oscillations (sorry if I missed this). It could also be possible to compare the strength of this ultrasensitivity to the previously proposed 'ultrasensitivity in KaiB-dependent KaiA sequestration that arises from opposing S and T phosphorylations 524 within hexamers (Lin et al., 2014)', or the transcriptional feedback loop due to KaiC regulating its own promoter. It is great that the authors find that the addition of ultrasensitivity to a previously generated model can increase the range of conditions where oscillations are possible, which is a step in this direction.

The ultrasensitivity in KaiC phosphorylation that we identified in this work is by itself insufficient to generate oscillation because it does not form a nonlinear delayed negative feedback loop (Novák and Tyson, 2008), in contrast to sequestration of KaiA by KaiBC complexes. In the latter process, the

cooperative binding of KaiB provides the nonlinear response, the ordering of phosphorylation and KaiB fold switch provide the delay, and the KaiB-mediated KaiA sequestration provides the negative feedback. The ultrasensitivity in KaiC phosphorylation, mediated by the interaction between KaiA and U-KaiC, results from mutual inhibition that effectively provides positive feedback on KaiA's activity; this introduces further nonlinearity but is not delayed.

The main role of the ultrasensitivity in KaiC phosphorylation is to stabilize the oscillation across %ATP conditions, because the threshold depends on %ATP. As noted in the last paragraph of Results and Fig. 4F, the capacity of KaiB (represented by $[C_{max}^{S+D}]$) to sequester KaiA is reduced at low %ATP, and the ultrasensitivity in KaiC phosphorylation compensates to stabilize the period. We have added an additional sentence in the Discussion section to clarify the roles of cooperative KaiB binding and ultrasensitivity in KaiC phosphorylation (new text highlighted in blue).

The reviewer also mentioned that the transcriptional feedback loop plays a role in generating oscillations *in vivo*. This is an interesting issue, and the architecture of this feedback loop is likely related to the need for the oscillator to function properly in a growing cell. However, a satisfactory analysis of this issue is outside the scope of the current manuscript, which focuses on properties that are intrinsic to the core oscillator.

Reviewer #2:

-Summary

Hong et al. analyzed the KaiC phosphorylation kinetics based on the Bayesian modeling and proposed a model to explain the robust oscillation against the ATP/ADP alteration, which is called the metabolic compensation. Ever since the reconstitution of the circadian rhythm of KaiC phosphorylation *in vitro*, the mathematical modeling approach has been applied to understand the robust oscillation of KaiC phosphorylation dynamics. In this study, the authors performed the data-driven approach instead of the forward modeling approach. Although KaiC protein is a homo-hexamers and each monomer has CI and CII domains, they modeled only a CII domain to enable the data-driven approach. After constructing the biochemical model of the CII domain, they sampled biochemical parameters to fit the biochemical experiments. From the sampled parameters, they found the unphosphorylated CII-ADP complex has a high affinity to KaiA. The authors also found the competition of the binding between different-KaiC states and KaiA generates the ultrasensitivity of the KaiC phosphorylation with respect to the KaiA concentration. The mechanism underlying the ultrasensitivity can be categorized as a substrate competition model. Based on these findings, they introduced the ultrasensitivity to the simplified oscillation model previously developed by the authors. By the modified mathematical model, the authors confirmed that the introduced ultrasensitivity increases the robustness of the oscillation by changing the ATP/ADP ratio.

-General remarks

The description of the Bayesian modeling is detailed and the model is well fitted to the experimental data. The computation of the Bayes factor for each model is useful to compare different models in a quantitative way. Thus, the current study has an impact on quantitative modeling in the field of computational biochemistry. The finding of ultrasensitivity in the action of KaiA to KaiC and the

mechanistic insight underlying it are important to the circadian field as well as general biochemistry. However, I have some issues to be addressed as follows:

-Major points

1. The authors write the section title "Substrate competition underlies metabolic compensation" in line 417 and the title "Ultrasensitivity Underlying Metabolic Compensation". This seems to be the overstatement because the authors do not perform any experiments to validate the mathematical model in Fig. 4. To state "underlies", the author should perform the verification experiment. In Fig. 3F, the authors showed KaiC-AA mutant can mimic the substrate competition by the unphosphorylated KaiC. Is the metabolic compensation affected or invariant by titrating KaiC-AA or KaiC-EE mutants in the in vitro oscillation experiment? Because the authors interpret the ultrasensitivity as the synchronization step, it should be also interesting to observe the amplitude change of reconstituted oscillation in the presence of KaiC-AA.

Per the reviewer's suggestion, we performed additional experiments to test the effect of KaiC-AA on the sensitivity of the clock to %ATP. The results of the experiment are shown in Fig. EV3C, and a brief discussion of the experiment is added to the section "A substrate competition mechanism underlies ultrasensitivity in KaiC phosphorylation" (new text highlighted in blue).

We found that the presence of KaiC-AA made the amplitude of the oscillation smaller and the period more sensitive to %ATP, especially at low %ATP conditions. However, the result of this experiment is difficult to interpret because we expect competing effects from the introduction of KaiC-AA. On the one hand, as implied by our argument in the manuscript, the presence of KaiC-AA should have a synchronizing effect at subjective night and thus promote oscillation at low %ATP by further inhibiting unsequestered KaiA. On the other hand, as our original titration experiment (Fig. 3G) demonstrated, KaiC-AA can also inhibit the phosphorylation reaction, which has the effect of reducing the amplitude of the phosphorylation cycle. When the amplitude is already small at low %ATP, the presence of KaiC-AA can thus kill the reaction. In other words, because of the dual, conflicting effects of the phosphorylation threshold, this experiment neither supports nor refutes our hypothesis.

Overall, we agree with the reviewer that the experimental evidence directly supports the existence of a metabolite-dependent ultrasensitive response, and that the relation between substrate competition and metabolic compensation is a model inference. We have changed the title and adjusted our language throughout the paper to reflect this.

2. In Eq. (38), the ultrasensitivity is introduced by hand. Since the authors found the substrate competition mechanism generates the ultrasensitivity, it seems to be possible to derive the kinetic equation like Eq. (28). At least it is requested to discuss whether the strength of observed KaiC(AA)-EC50 relationship (Fig. 3G) is reasonable to the level of ultrasensitivity estimated by the authors' model.

Per the reviewer's suggestion, we attempted to derive an analytical expression for the steady-state phosphorylation level of KaiC, assuming a substrate competition mechanism similar to that by Ferrell and Ha (2014b) (i.e., Eq. 28 in the current manuscript). The kinetic model we consider is detailed in the diagram below:

Similar to the substrate competition mechanism by Ferrell and Ha, we consider the nucleotide exchange activity of KaiA on T-KaiC; that is, the portion of the diagram in the gray dotted box is mathematically equivalent to Fig. 8A in Ferrell and Ha (2014b). We extend the model by considering the (de)phosphorylation, hydrolysis, and nucleotide-exchange activity of U-KaiC.

We attempted to solve for the steady-state solution of the model and derive an expression for the %phosphorylation level of KaiC. The resulting expression is complicated (see attached Mathematica notebook), and we found it difficult to extract an intuitive interpretation from it. Given these results, we find it preferable to introduce ultrasensitivity by hand as we did in Eq. 38, where a phosphorylation threshold is introduced that depends on both %ATP (as predicted by the full model in Fig. 3A) and [U-KaiC] (as quantified in Fig. 3G).

To address the second part of the reviewer's concern, we simulated the KaiC-AA titration experiment using the modified Phong model and compared the results with experiment (Fig. EV3E). Additional discussion of this analysis is added to the Appendix (section "Phenomenological modifications to the Phong model"; last paragraph). In brief, we found that the scaling of EC50 with KaiC-AA in the model is consistent with the experimental results, but the EC50 in the model is consistently larger than the experiment by about 0.15 μ M. This suggests that the level of ultrasensitivity we added to the Phong model is reasonable, given the phenomenological nature of both the model and the way ultrasensitivity is added.

-Minor points

1. In Fig. 1D, dissociation constants are denoted as K_d^{DP} or K_d^{TP} . This notation cannot discriminate the phosphorylation states. Since the authors use the notation $k_a\{U,DP\}$ for the rate constant, it seems to be better to use $K_d\{X,DP\}$ instead of K_d^{DP} . Then, it is helpful to show labels such as $X = U, T, S, D$ with corresponding colors in the inset of the figure.

The figure and its legend are now updated to incorporate the reviewer's suggestion.

2. In Fig. 3E, the labels "threshold" and "switch" are ambiguous. They should be replaced with EC10 and EC90 - EC10, respectively.

The axes labels are now updated according to the reviewer's suggestion.

3. In Fig. 4G, what is represented by orange-rectangles in the background?

The yellow background corresponds to time periods when $[\text{KaiA}]_{\text{active}}$ (gray curve) drops below EC50 (green curve). However, we have since deleted Fig. 4G from the manuscript because we felt that it did not effectively convey the idea that the size of the phosphorylation limit cycle scales with %ATP.

4. In Fig. S1B, S4DF, and S10B, labels, and units of the x-axis are missing.

The horizontal axes in these figures (now Figs. EV1B and EV2DF) are now labeled. We note that the multiplicative factors in Fig. S10B (now Fig. S5) are dimensionless.

5. In Fig. S1B, distributions of $k_{\text{TP}}^{\text{A,X}}$ are presented in "Nuc. exch.". To my understand, these parameters depend on %ATP as $k_{\text{r}}^{\text{DP,X}} * [\text{ATP}] / ([\text{ATP}] + [\text{ADP}])$ from Eq. (4). If so, %ATP-independent parameters $k_{\text{r}}^{\text{DP,X}}$ should be plotted instead of $k_{\text{TP}}^{\text{A,X}}$.

In the original figure (now Fig. EV1B), the distributions of $k_{\text{r}}^{\text{DP,X}}$ were plotted assuming that there is 100% ATP (i.e., $k_{\text{r}}^{\text{DP,X}} = k_{\text{TP}}^{\text{A,X}}$). We have updated the parameter names to $k_{\text{TP}}^{\text{A,X}}$ according to the reviewer's suggestion to clarify the issue.

6. In the legend of Fig. S4, it is helpful to cite references of the original experimental data.

The references are now added to the figure legend (now Fig. EV2).

7. In Fig. S6, the notation $(dC/C)/(dk/k)$ in x and y labels is not correct. It should be replaced with $d\log(C)/d\log(k)$ or $(dC/dk)/(C/k)$.

The axes labels in the figure are now corrected (now Fig. S4).

8. Since Fig. S7A is a different representation of Fig. 2D, Fig. S7A can be removed. Then, it seems to be readable to move Fig. S7B to the next of Fig. 2D with the scatter plot similar to Fig. 2D.

We removed Fig. S7A (Fig. S7 is now Fig. EV3) and replaced Fig. S7B with a scatter plot (which is now Fig. EV3A). However, because the discussion of the plate reader experiment is primarily centered on the sensitivity of the period (rather than amplitude) on %ATP and [KaiA], we believe that Fig. EV3A is more appropriate as an EV figure panel rather than a main figure panel.

9. Typo in line 824. Aligent should be Agilent.

This typographical error is corrected.

10. In line 847, the authors wrote "(i.e., $[\text{ATP}] + [\text{ADP}]$) was held constant at 5 uM.". I guess the unit of concentration is not uM but mM.

The unit is indeed mM, and this typographical error is corrected.

11. In Table 4, it is better to show the absolute concentrations of ATP and ADP instead of %ATP.

Table 4 is now updated according to the reviewer suggestion.

12. I could not find the numerical method to solve the modified Phong model and the analysis method to compute the period shown in Fig. 4E. It is appropriate to describe the method in or after the section "Phenomenological modifications to the Phong model" (line 1069).

A paragraph on the numerical method for the Phong model is now added to the computational methods section (new text highlighted in blue).

Reviewer #3:

Here, the authors use a Bayesian parameter estimation approach to uncover metabolic mechanisms underlying the cyanobacterial circadian clock. Specifically, they investigate the daytime interactions among the various phosphorylation and nucleotide-bound states of KaiC and KaiA. First, they use the results from parameter estimation to illustrate the transient kinetic states of the KaiC protein. In particular, their model replicates the KaiC preference of phosphorylation at the T site before phosphorylation at the S site. Using their model, they propose that the preference is more than just a difference in the relative unphosphorylated to T-phosphorylated and unphosphorylated to S-phosphorylated reaction rates. They also show that the ADP-bound states of KaiC are unstable relative to the ATP-bound states. Next, they show that the KaiA concentration, to an extent, mitigates the clock's sensitivity and sensitivity of the period to the percent of ATP present. Their analysis reveals, however, that KaiA concentration is not sufficient to fully abolish the sensitivity of the clock to percent ATP.

The remaining results discuss the ultrasensitivity of KaiC phosphorylation on KaiA levels. The authors appeal to previous work on ultrasensitivity to justify their claim that KaiC phosphorylation is ultrasensitive to KaiA binding affinity to the nucleotide state of unphosphorylated KaiC. See below for two main criticisms regarding this conclusion and how the authors can address them. The authors argue that the mechanism is substrate competition, where the competition is driven by differential binding to multiple substrates. They corroborate this hypothesis through two experiments. First, they measure the stimulus-response relation of KaiC S431A mutant in the presence of both KaiC S431A/T432A and KaiCS431E/T432E. Each mutant acts as a competitor to KaiA-KaiC interaction. Second, they show that the model exhibits weak nonlinearity when KaiA has near equivalent binding affinity for the two nucleotide-bound states. To show this, they use a Bayes factor calculation revealing that near equivalent binding affinity significantly reduces the quality of the fit. In contrast, a model with differential binding affinities for the nucleotide-bound state with equivalent affinity across phosphorylation states has little effect on the fit quality under the Bayes factor.

Finally, the authors show that ultrasensitivity leads to metabolic compensation of the full oscillator. Previous models of the KaiABC oscillator require a high binding affinity between KaiB and KaiA, leading to sequestration of KaiA, to exhibit oscillations. However, the authors show that coupling the protein sequestration with the ultrasensitivity leads to more robust oscillations. They provide experimental evidence that protein sequestration is not solely responsible for oscillations and give a compelling description of how the ultrasensitivity underlies metabolic compensation.

The cyanobacterial is a well-studied model organism in the field of chronobiology with many of the mechanisms underlying the core clock well understood. The authors advance the field using *in silico* and experimental methods to uncover an important aspect to the generation of oscillations in cyanobacteria, namely the ultrasensitivity of KaiC phosphorylation to differential binding affinities of KaiA. The result is significant in that it challenges the current view that sequestration of KaiA by KaiBC provides sufficient nonlinearity in the system to generate oscillations. The study will be of interest to both circadian and computational biologists.

Major Concerns

1) While I like the Bayesian modeling approach and the data specifically collected to test the model, I think the manuscript overemphasized this approach in that I don't think the manuscript will be read because of the Bayesian approach. In my mind, what the manuscript says about the mechanism of cyanobacterial timekeeping is of sufficient interest to readers, and the approach they use is not that different than what is found in some other models. So, while I think it is good to highlight the approach as a strength of the manuscript, it seems overemphasized.

By fitting a mechanistically agnostic model to data within a Bayesian framework, we are able to systematically discover mechanistic features (i.e., ultrasensitivity and substrate competition) and quantify their uncertainties. This would not be straightforward with a traditional approach that examined the behavior of a phenomenological model formulated based on previous observations. Because the traditional approach still predominates in the field, we feel that the methods employed in our paper will be of interest to researchers working on systems beyond the circadian clock. We thus want to highlight the approach. That said, we have revised the Introduction significantly to address the reviewer's comment.

2) The reader has to wade through quite a number of details until the most interesting aspects of the manuscript (e.g., about sequestration) are presented. I think that some reorganization or rewording could make the manuscript more approachable to a general audience. In general, the writing is good, but I do think that, written the right way, the main biological takeaways could be more easily apparent. At the moment the manuscript reads more as would a chronological description of what happened and I think jumping in the text to some more of the key points may be helpful.

The manuscript develops the model and presents its basic behavior, which is consistent with existing knowledge, prior to the unanticipated findings. While we understand the reviewer's desire to get to the most interesting points as quickly as possible, we tried a number of orderings when drafting the paper and felt that the current layout was the only one that best enabled readers to understand the proposed mechanism and the significance of the ultrasensitivity (i.e., substrate competition). We've revised the last three paragraphs of the Introduction to provide a clear roadmap to the main results from the paper, including a summary of the main biological takeaways, and we have put significant effort into titling subsections; acknowledging the reviewer's concern, we added a paragraph near the start of the Results section stating that, depending on readers' interests, the first two subsections can be skipped without loss of continuity (new text highlighted in blue). We believe that this should enable readers familiar with the system to skip ahead to the most interesting results without sacrificing accessibility to readers less familiar with the system.

3) One clarification concerns the conclusion of the ultrasensitivity dependence of KaiC phosphorylation on KaiA levels. The authors use the phrase "threshold-hyperbolic stimulus-response relation" and cite the work of Gomez-Urbe from 2007. In that work, however, the authors identify four distinct responses of steady-state response. The third response is a "threshold-hyperbolic stimulus-response function", which is a different response than the fourth response, ultrasensitivity.

As the reviewer points out, the authors of Gomez-Urbe *et al.* (2007) identified four types of stimulus-response relations. In this scheme a threshold-hyperbolic response curve is indeed not considered "ultrasensitive," which is reserved for sigmoidal response curves exclusively. However, in the literature

there are varying uses of the term “ultrasensitive.” For example, the original Goldbeter-Koshland definition was any input-output relationship with $EC_{90}/EC_{10} < 81$, so it would encompass the different types of stimulus-response relations considered by Gomez-Uribe et al.; see Ferrell and Ha (2014a) for a recent review. We thus use “ultrasensitive” in a broader sense to refer to any input-output function that saturates but is nonlinear for small values of the input. We use “threshold hyperbolic” when we specifically characterize the shape of the function and cite Gomez-Uribe *et al.* accordingly.

4) Next, the authors conclude ultrasensitivity using a metric from previous work by Gunawardena. In particular, they calculate $EC_{90}-EC_{10}$ where EC_{90} (EC_{10}) is the KaiA concentration required to reach 90% (respectively, 10%) of the steady-state phosphorylation level at saturation. However, in his work from 2005, Gunawardena uses the cooperativity index defined as EC_{90}/EC_{10} . The value EC_{90}/EC_{10} was used in the original work on ultrasensitivity by Goldbeter and is the canonical measure of ultrasensitivity still. The authors should normalize their response curves and recalculate their cooperativity index using EC_{90}/EC_{10} and compare the values to that of the defined bound of ultrasensitivity, namely 81. Or, at least, they should note that their metric is different from that of Gunawardena and justify its use.

Gunawardena (2005) defines two metrics on p. 14621 (second paragraph):

$$\tau_{\epsilon}(f) = f^{-1}(\epsilon) \quad \text{and} \quad \sigma_{\epsilon}(f) = \theta_{\epsilon}(f) - \tau_{\epsilon}(f)$$

for a sigmoidal dose-response curve f , where $\theta_{\epsilon}(f) = f^{-1}(1 - \epsilon)$. Written in our notation, $EC_{10} = \tau_{0.1}(f)$ and $EC_{90} - EC_{10} = \sigma_{0.1}(f)$. Thus the use of the difference is consistent with Gunawardena (2005).

5) Mass action should be hyphenated when used as an adjective (as in mass-action kinetics on lines 140, 351, and in the caption of figure 1).

All occurrences of the phrase “mass action kinetics” in the manuscript have been updated to “mass-action kinetics”.

6) It would be nice near the derivation of Eqn. (36) in the Supplemental Information to point out its similarities to the work of Kim and Forger, 2012.

The reference is added above Eqn. 36.

7) The authors italicize apo in the Supplemental Information but not in the main text on line 167.

The word “apo” is now un-italicized throughout the manuscript.

9th Apr 2020

Manuscript Number: MSB-19-9355R

Title: Bayesian Modeling Reveals Ultrasensitivity Underlying Metabolic Compensation in the Cyanobacterial Circadian Clock

Author: Lu Hong

Danylo Lavrentovich

Archana Chavan

Eugene Leypunskiy

Eileen Li

Charles Matthews

Andy LiWang

Michael Rust

Aaron Dinner

Dear Prof Dinner,

Thank you for sending us your revised manuscript. We have now heard back from the three reviewers who agreed to evaluate your manuscript. You will see from the comments below that all reviewers are overall positive and support publication of the article in *Molecular Systems Biology*. I am pleased to inform you that your manuscript will be accepted in principle pending the following essential amendments:

1. Please address reviewer #1's concern by improving the introduction in light of previously published work.

On a more editorial level:

1. Please provide a .doc formatted version of the manuscript text (including Figure legends and tables).

2. A Conflict of Interest statement should be provided in the main text.

3. Please remove the synopsis text from the main manuscript.

4. Please note that we now mandate that all corresponding authors list an ORCID digital identifier (one is missing now). This takes less than 90 seconds to complete. We encourage all authors to supply an ORCID identifier, which will be linked to their name for unambiguous name identification.

5. I have only slightly modified the synopsis text. Could you let me know if you would like to introduce further modifications?

This study takes a data-driven kinetic modeling approach to characterize the interaction between KaiA and KaiC in the cyanobacterial circadian clock to understand how the oscillator responds to changes in cellular metabolic conditions.

- An extensive dataset of KaiC autophosphorylation measurements is generated and used to constrain a detailed yet mechanistically naive kinetic model within a Bayesian parameter estimation

framework.

- KaiA concentration tunes the sensitivity of KaiC autophosphorylation and the period of the full oscillator to %ATP.
- The model reveals an ultrasensitive dependence of KaiC phosphorylation on KaiA concentration as a result of differential KaiA binding affinity to ADP- vs. ATP-bound KaiC.
- Ultrasensitivity in KaiC phosphorylation likely contributes to metabolic compensation by suppressing premature phosphorylation at nighttime.

When you resubmit your manuscript, please download our CHECKLIST (<http://bit.ly/EMBOPressAuthorChecklist>) and include the completed form in your submission. *Please note* that the Author Checklist will be published alongside the paper as part of the transparent process (<https://www.embopress.org/page/journal/17444292/authorguide#transparentprocess>)

Click on the link below to submit your revised paper.

Link Not Available

Please submit your revised manuscript within two weeks. I look forward to seeing a revised version of your manuscript soon.

Yours sincerely,
Jingyi Hou

Jingyi Hou
Editor
Molecular Systems Biology

If you do choose to resubmit, please click on the link below to submit the revision online before 9th May 2020.

Link Not Available

IMPORTANT: When you send your revision, we will require the following items:

1. the manuscript text in LaTeX, RTF or MS Word format
2. a letter with a detailed description of the changes made in response to the referees. Please specify clearly the exact places in the text (pages and paragraphs) where each change has been made in response to each specific comment given
3. three to four 'bullet points' highlighting the main findings of your study
4. a short 'blurb' text summarizing in two sentences the study (max. 250 characters)
5. a 'thumbnail image' (550px width and max 400px height, Illustrator, PowerPoint or jpeg format),

which can be used as 'visual title' for the synopsis section of your paper.

6. Please include an author contributions statement after the Acknowledgements section (see <https://www.embopress.org/page/journal/17444292/authorguide#manuscriptpreparation>)

7. Please complete the CHECKLIST available at (<http://bit.ly/EMBOPressAuthorChecklist>). Please note that the Author Checklist will be published alongside the paper as part of the transparent process

(<https://www.embopress.org/page/journal/17444292/authorguide#transparentprocess>).

8. Please note that corresponding authors are required to supply an ORCID ID for their name upon submission of a revised manuscript (EMBO Press signed a joint statement to encourage ORCID adoption) (<https://www.embopress.org/page/journal/17444292/authorguide#editorialprocess>).

Currently, our records indicate that the ORCID for your account is 0000-0001-8328-6427.

Link Not Available

The system will prompt you to fill in your funding and payment information. This will allow Wiley to send you a quote for the article processing charge (APC) in case of acceptance. This quote takes into account any reduction or fee waivers that you may be eligible for. Authors do not need to pay any fees before their manuscript is accepted and transferred to the publisher.

*** PLEASE NOTE *** As part of the EMBO Press transparent editorial process initiative (see our Editorial at <http://dx.doi.org/10.1038/msb.2010.72> , Molecular Systems Biology will publish online a Review Process File to accompany accepted manuscripts. When preparing your letter of response, please be aware that in the event of acceptance, your cover letter/point-by-point document will be included as part of this File, which will be available to the scientific community. More information about this initiative is available in our Instructions to Authors. If you have any questions about this initiative, please contact the editorial office (msb@embo.org).

Reviewer #1:

I am happy that the authors have addressed my points.

One last point I have is that there is some literature on Bayesian modelling on circadian clocks that should be briefly discussed in the introduction. Currently the authors have written:

'While Bayesian parameter estimation (MacKay and Kay, 2003) has been used occasionally in systems biology (Flaherty et al., 2008; Klinke, 2009; Toni et al., 2009; Xu et al., 2010; Schmidl et al., 2012; Eydgahi et al., 2013; Pullen and Morris, 2014; Mello et al., 2018), here it is particularly useful; it allows us to estimate parameter values, quantify the importance of specific model elements, and make mechanistic predictions from the model.'

A reader might get the impression that Bayesian modelling has not been applied to circadian clocks before. There are a few examples where Bayesian modelling has been used, for example for parameter optimisation in the plant circadian clock (eg. PMID: 24267177), or estimation of the coupling of the clock to the cell cycle in cyanobacteria (PMID: 30409801) or for inferring oscillatory network structures (PMID: 26177966). It would help if these were mentioned in the introduction. None of these references affect the novelty of the current work.

Reviewer #2:

The authors have done an additional experimental verification and re-evaluation of the model parameters. I understand the limitation of the use of the KaiC-AA mutant to perturb the substrate-competition and synchronization steps. The authors amended the title and other parts of the manuscript adequately, and this study is now ready for publication.

Reviewer #3:

The authors have addressed my concerns. I have no further concerns and think this is a valuable addition to the literature.

24th Apr 2020

Manuscript number: MSB-19-9355RR

Title: Bayesian modeling reveals metabolite-dependent ultrasensitivity in the cyanobacterial circadian clock

Dear Prof Dinner,

Thank you again for sending us your revised manuscript. We are now satisfied with the modifications made and I am pleased to inform you that your paper has been accepted for publication.

*** PLEASE NOTE *** As part of the EMBO Publications transparent editorial process initiative (see our Editorial at <http://dx.doi.org/10.1038/msb.2010.72>), Molecular Systems Biology publishes online a Review Process File with each accepted manuscripts. This file will be published in conjunction with your paper and will include the anonymous referee reports, your point- by-point response and all pertinent correspondence relating to the manuscript. If you do NOT want this File to be published, please inform the editorial office at msb@embo.org within 14 days upon receipt of the present letter.

Should you be planning a Press Release on your article, please get in contact with msb@wiley.com as early as possible, in order to coordinate publication and release dates.

LICENSE AND PAYMENT:

All articles published in Molecular Systems Biology are fully open access: immediately and freely available to read, download and share.

Molecular Systems Biology charges an article processing charge (APC) to cover the publication costs. You, as the corresponding author for this manuscript, should have already received a quote with the article processing fee separately.

Please let us know in case this quote has not been received.

Once your article is at Wiley for editorial production you will receive an email from Wiley's Author Services system, which will ask you to log in and will present you with the publication license form for completion. Within the same system the publication fee can be paid by credit card, an invoice or pro forma can be requested.

Payment of the publication charge and the signed Open Access Agreement form must be received before the article can be published online.

Molecular Systems Biology articles are published under the Creative Commons licence CC BY, which facilitates the sharing of scientific information by reducing legal barriers, while mandating attribution of the source in accordance to standard scholarly practice.

Proofs will be forwarded to you within the next 2-3 weeks.

Thank you very much for submitting your work to Molecular Systems Biology.

Sincerely,

Jingyi Hou
Editor
Molecular Systems Biology

Corresponding Author Name: MJ RUST & AR DINNER

Journal Submitted to: MSB

Manuscript Number: MSB-19-9355